# Probing the thermal decomposition mechanism of CF$_3$SO$_2$F by deep learning molecular dynamics
Anyang Wang [1], Zeyuan Li[2], Shubo Ren[1], Xue Ke[1], Xuhao Wan[1], Rong Han[1], Xianglian Yan[3], Wen Wang[3], Yu Zheng [1] ✉, Yuzheng Guo [1,2] ✉ & Jun Wang [1] ✉

The urgent need to phase out SF$_6$, an extremely potent greenhouse gas prevalent in electrical grids, drives the search for eco-friendly insulation alternatives. Trifluoromethanesulfonyl fluoride (CF$_3$SO$_2$F) emerges as a promising candidate due to its excellent properties. However, understanding its thermal decomposition pathways and products under operationally relevant conditions is critical for evaluating its environmental feasibility and mitigating potential risks upon accidental release or during fault events. This study investigates the thermal decomposition mechanisms of CF$_3$SO$_2$F using a deep learning potential that combines ab initio accuracy with empirical MD efficiency. By leveraging machine learning driven molecular dynamics, we systematically analyze the yields and components of decomposition products versus temperatures, gas mixing ratios, and buffer gas. The results reveal that the bond-breaking pathways are temperature-dependent, with both elevated temperatures and higher buffer gas mixing ratios promoting its decomposition. Elevated gas pressure enhances the decomposition process by increasing the collision frequency among reactant species. Additionally, N$_2$ exhibits an inhibitory effect on decomposition under high pressure compared to CO$_2$. Experimental validation via a thermal decomposition platform confirms characteristic decomposition products. These findings are pivotal for guiding the rational design and safe deployment of CF$_3$SO$_2$F to achieve substantial greenhouse gas mitigation in the power industry.

Insulating gases play an irreplaceable role in power systems. Sulfur hexa-fluoride (SF$_6$) has been widely used as an insulating gas in power equipment such as switchgear and transmission pipelines due to its excellent insulation and arc extinguishing properties[1,2]. However, the potential greenhouse effect caused by high global warming potential (GWP = 23900) and atmospheric lifetime (3200 years) of SF$_6$ will have an irreversible impact on the atmosphere and the earth's environment, and it is imperative to explore the green and eco-friendly new insulation medium[3–6]. Researchers have done much work in SF$_6$ alternative gases[7,8]. So far, some gases with comparatively high insulating properties, such as CF$_3$I, C$_5$F$_{10}$O, and C$_4$F$_7$N, and their mixtures are being investigated as SF$_6$ replacement gases[9–11]. However, they have different defects in dielectric strength, liquefaction temperature, GWP, toxicity, stability and so on. Even C$_4$F$_7$N, the most popular gas in SF$_6$ substitution research, faces the challenge of high liquefaction temperature, making its application in alpine areas difficult[12].

In recent years, trifluoromethylsulfonyl fluoride (CF$_3$SO$_2$F) has proven to be an eco-friendly insulation replacement gas with excellent potential and performance. Not only the AC and DC breakdown voltage of CF$_3$SO$_2$F can be increased to 1.3–1.6 times that of SF$_6$ under the same conditions, but also the GWP of CF$_3$SO$_2$F (3678) is much lower than that of SF$_6$[13,14]. In addition, the lower liquefaction temperature (–22 °C), excellent gas-solid compat-ibility, low toxicity, and high stability of CF$_3$SO$_2$F have been proven[15,16]. However, gaseous dielectrics can decompose under discharge or localized overheating faults, forming inevitable byproducts. The study of the decomposition characteristics of gas-insulating media is an essential indi-cator of its reliability, closely related to its self-recovery and insulation performance[17–19]. Wang et al. analyzed the possible decomposition path-ways and byproducts of CF$_3$SO$_2$F by quantum chemical methods[15]. Our team calculated the equilibrium composition, thermodynamic properties, and transport coefficients of CF$_3$SO$_2$F gas mixtures and found that the gas

[1]School of Electrical Engineering and Automation, Wuhan University, Wuhan, China. [2]School of Power and Mechanical Engineering, Wuhan University, Wuhan, China. [3]High Voltage Department, China Electric Power Research Institute, Beijing, China. ✉e-mail: zywhuee@whu.edu.cn; yguo@whu.edu.cn; junwangwhu@whu.edu.cn

mixing ratio and pressure affect the thermophysical and transport properties of $CF_3SO_2F$ gas mixtures[20,21]. Despite these contributions, current kinetic and thermodynamic studies present fundamental limitations. Existing research often overlooks the dynamic decomposition properties of $CF_3SO_2F$, especially under varying conditions, which impedes risk assessment and safe deployment[22]. The experimental difficulties and high costs associated with such studies exacerbate these gaps, resulting in a lack of comprehensive insights into the micro-mechanisms underlying these processes[23,24].

Theoretical approaches are powerful tools to study the dynamic decomposition properties of $CF_3SO_2F$ mixtures at the atomic level[15,25]. The molecular simulation community has long faced the problem of accuracy and efficiency in modeling potential energy surfaces and interatomic forces. For instance, although the ab initio molecular dynamics (AIMD) simulations based on density functional theory (DFT) could show sufficient accuracy in describing chemical reactions that contain bond cleavage and formation, its computational cost of such high-level methods would limits their application to systems with hundreds to thousands of atoms[26,27]. Conversely, empirical and reactive force fields allow for larger and longer simulations[24,28–31]. However, parameters fitting in these methods are usually lengthy processes, and their accuracy and transferability are often questioned. Therefore, a theoretical approach with quantum chemical accuracy and lower computational resource requirements is needed for high-precision simulations of the dynamic thermal decomposition process of $CF_3SO_2F$ gas mixtures. Over the past few years, machine learning methods using DFT data have achieved some notable successes in the characterization of molecular macrosystems[32–34]. Deep learning potential (DLP) developed on machine learning can automatically extract features from DFT data for deep neural network training and achieve preset accuracies[35,36]. DLP can take into account the time cost of empirical force fields while maintaining accuracy in ab initio. Yang et al. have successfully used DLP to simulate the dynamic and complex decomposition process of urea in water[37].

Inspired by machine learning-based approaches to molecular simulations, a neural network-based machine learning potential with comparable accuracy to ab initio and comparable efficiency to empirical potential-based molecular dynamics was developed to describe the dynamic thermal decomposition of $CF_3SO_2F$ mixtures. Based on the machine-learning potential-driven molecular dynamics (MD) simulations, the decomposition

mechanism of $CF_3SO_2F$ mixtures at different temperatures and gas mixing ratios and components of the main decomposition products were obtained. The effects of the $CF_3SO_2F$ mixing ratio on the decomposition of the $CF_3SO_2F$ mixture were analyzed to reveal the mechanism of the thermal decomposition of the $CF_3SO_2F$ mixture under different pressures. Compared with $CO_2$, $N_2$ as a buffer gas can inhibit the decomposition of $CF_3SO_2F$ to a certain extent. The thermal decomposition properties of $CF_3SO_2F$ gas mixtures were experimentally investigated using a constructed thermal decomposition platform, and characteristic decomposition products aiming to characterize the failure were proposed. This work provides significant insight into the thermal decomposition behavior of $CF_3SO_2F$ and its mixtures, contributing to the understanding of its stability and decomposition pathways under extreme conditions. Furthermore, the computational framework is potentially transferable for investigating the physicochemical properties and decomposition mechanisms of other promising insulating gases, guiding the development of eco-friendly alternatives to $SF_6$.

## Results

### DLP training and structure of simulation boxes

Figure 1 illustrates the workflow involved in training DLP and calculating the decomposition products of $CF_3SO_2F$ at various temperatures based on DLP. Initially, a potential is trained on a dataset composed of structures randomly selected from the AIMD reaction trajectories of a model with well-determined elemental and molecular configurations. This initial potential serves as a starting point for an iterative training process. Subsequently, the DLP of $CF_3SO_2F$ is developed and trained using potentials, forces, and virials from the AIMD dataset, which consists of a temperature range from 300 K to 3200 K in the NVT ensemble[35]. Through multiple explorations, labeling, and training, we compiled a comprehensive dataset of the $CF_3SO_2F$ gas mixture configurations across a wide range of temperatures[27,38]. The DLP was trained and fine-tuned after the dataset was built, and extensive validation tests confirmed that it would produce only small errors. Detailed information on the settings used for AIMD computation and DLP training is given in the methods section.

The reliability of the deep learning potential for modeling reactive events was rigorously validated. Initially, a machine-learning potential was trained on a dataset constructed by systematically sampling structures at

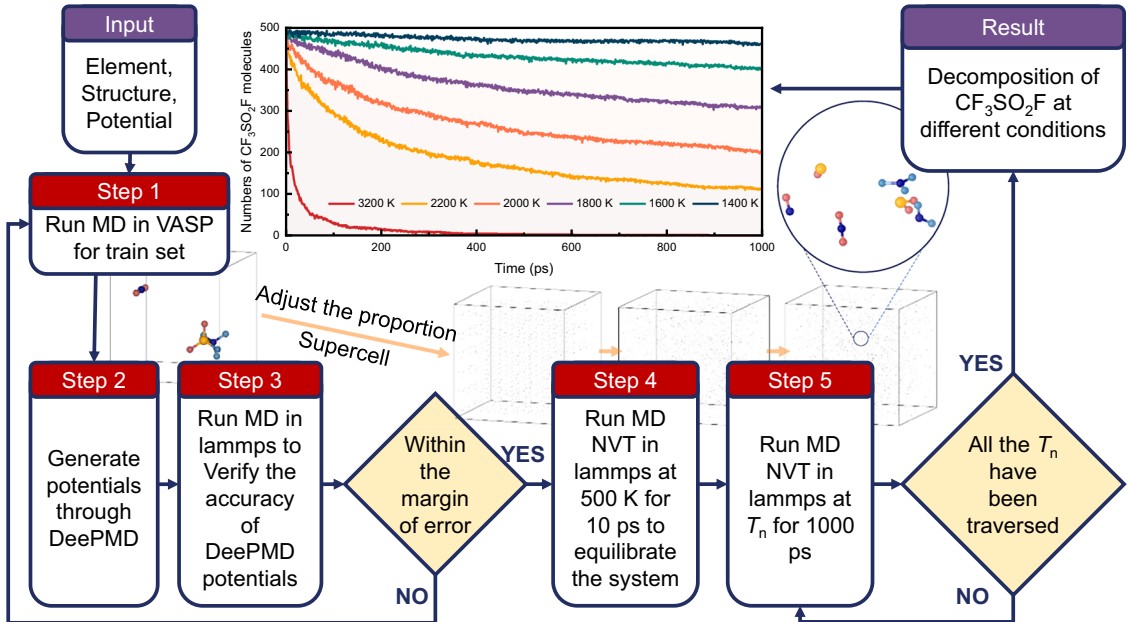

**Fig. 1 | Workflow for constructing and validating the deep learning potential for $CF_3SO_2F$ decomposition.** The workflow comprises four key steps: (1) Configuration sampling from AIMD trajectories, (2) Training of the DLP, (3) Validation of the DLP against DFT benchmarks, and (4) Large-scale molecular dynamics simulations of $CF_3SO_2F$ decomposition across a range of temperatures ($T_1$ to $T_n$), with supercell size adjusted to model different system pressures.

**Table 1 | Parameters of the simulated $CF_3SO_2F/CO_2$ systems**

| No | P | mixing ratio | $CF_3SO_2F$ | $CO_2$ | number of atoms | Density (g/cm3) | box length/Å |
|---|---|---|---|---|---|---|---|
| 1 | 0.1 MPa | 0.20 | 500 | 2000 | 10000 | 0.002681 | 466.6134 |
| 2 | 0.1 MPa | 0.14 | 500 | 3071 | 13213 | 0.002417 | 525.5068 |
| 3 | 0.1 MPa | 0.12 | 500 | 3667 | 15001 | 0.002328 | 553.2428 |
| 4 | 0.3 MPa | 0.12 | 500 | 3667 | 15001 | 0.006985 | 383.5971 |
| 5 | 0.5 MPa | 0.12 | 500 | 3667 | 15001 | 0.011642 | 323.5384 |
| 6 | 0.1 MPa | 0.14 | 583 | 3584 | 15416 | 0.002681 | 553.2428 |
| 7 | 0.1 MPa | 0.20 | 833 | 3334 | 16666 | 0.002417 | 553.2428 |

The pressure listed is the initial ideal gas pressure calculated from the given number of molecules in the simulation cell at 300 K.

regular time intervals from the AIMD reaction trajectories. This procedure ensures that the dataset captures a representative ensemble of the well-defined, chemically relevant configurations sampled during the reactive dynamics, covering both stable intermediates and transition regions, as shown in Fig. S1. The dataset is randomly divided into a training set (comprising 90% of the structures) and a validation set (comprising 10% of the structures). The training set was employed to fit a neural network, with internal validation performed using the validation set to ensure that the error on the validation set is not significantly higher than that on the training set. As illustrated in Fig. S2, the trained potential function accurately reproduces the DFT data, yielding a root-mean-square error (RMSE) of merely 1.3 meV for the total energy of each atom. The RMSE for the forces is 0.023 eV/Å, and the RMSE for the virials per atom is 2.5 meV, closely aligning with the DFT results. The deviation distribution plots for energy, force, and virials are all centered around zero, indicating the model's robust predictive capability.

To quantitatively assess the potential's accuracy, the potential energy surfaces of the $CF_3SO_2F$ molecule with changing $CF_3-SO_2F$, $F-CF_2SO_2F$, $F-CF_3SO_2$, and $O-CF_3SOF$ bond lengths were computed with both the DLP and the reference DFT method, showing excellent agreement (Fig. S3). Moreover, the DLP's description of key stationary points was consistent with high-level CCSD(T) reference data from the literature[15]. These results consistently identify C–S bond homolysis as the dominant initial decomposition step, followed by dissociation of the resulting radical, which confirms its transferability and robustness for probing the decomposition mechanism. The principal advantage of the DLP approach lies in its capacity to go beyond static energy calculations and provide statistically meaningful insights into finite-temperature reaction dynamics, including product branching ratios, at a computational cost inaccessible to direct ab initio molecular dynamics. Finally, the trained DLP was utilized to simulate the $CF_3SO_2F/CO_2$ gas mixture (hereafter denoting a mixture of $CF_3SO_2F$ in a $CO_2$ buffer gas) at various temperatures.

Molecular dynamics simulations using the trained DLP were conducted with LAMMPS[39]. Seven initial simulation boxes under different states, which contain at least 10,000 atoms to simulate the real environment, were created by randomly placing the molecules in a cubic box, as detailed in Table 1. For instance, simulation box No. 3 comprises 500 $CF_3SO_2F$ molecules and 3667 $CO_2$ molecules, totaling 15,001 atoms, with a cubic box side length of 553.24 Å and a 0.002328 g/cm³ density. This configuration corresponds to an actual condition of a 12% $CF_3SO_2F/CO_2$ gas mixture at 25 °C and 0.1 MPa.

It has been demonstrated that $CF_3SO_2F$ gas mixtures with less than 20% $CF_3SO_2F$ mixing ratio are more advantageous for engineering applications[40,41]. To investigate the decomposition of $CF_3SO_2F$ gas mixtures under different mixing ratios and the impact of these mixing ratios on decomposition, simulations were performed with varying $CF_3SO_2F$ mixing ratios. No.1 and No.2 simulation boxes maintain the total $CF_3SO_2F$ molecules fixed and simulate 20% $CF_3SO_2F/CO_2$ and 14% $CF_3SO_2F/CO_2$ systems, respectively. Similarly, No.6 and No.7 simulation boxes keep the total number of molecules constant to simulate the same systems. In addition, simulation boxes No. 4 and No. 5 represent the decomposition of 12%

$CF_3SO_2F/CO_2$ gas mixture at 0.3 MPa and 0.5 MPa to investigate the effect of pressure on decomposition respectively. The simulation boxes for $CF_3SO_2F/N_2$ (hereafter denoting a mixture of $CF_3SO_2F$ in a $N_2$ buffer gas) under different conditions were built according to the same approach as shown in Table S1.

## Decomposition of $CF_3SO_2F$

The two primary causes of decomposition in the insulating dielectric region are localized overheating faults and high temperatures resulting from localized (corona) or arc discharges[24]. The temperature in the core region of a localized discharge ranges from 700 to 1200 K, while in the arc discharge region, it spans from 3000 to 12000 K. To investigate the effect of temperature on the decomposition characteristics of the gas mixture, molecular dynamics simulations of the model were conducted in the range of 300 K to 3200 K. $CF_3SO_2F$ in $CF_3SO_2F/CO_2$ begins to decompose at 1400 K, indicating that it remains stable in the presence of localized discharges as shown in Fig. 2a.

The buffer gas $CO_2$ does not decompose between 1400 K and 2200 K, aligning with previous studies[24]. As the temperature increases from 1400 to 2200 K, the decomposition rate of $CF_3SO_2F$ rises, with the decomposition ratio (the percentage of decomposed $CF_3SO_2F$) within 1000 ps increasing from 0.07 to 0.77, as illustrated in Fig. 2d. When the temperature reaches 3200 K (Fig. 2a), the decomposition ratio of $CF_3SO_2F$ surges to 0.94 within 1000 ps, indicating rapid decomposition in the arc discharge channel.

We further investigate the effect of pressure on the decomposition behavior of $CF_3SO_2F$. The decomposition of $CF_3SO_2F$ under pressures of 0.1, 0.3, and 0.5 MPa at 2200 K is shown in Fig. 2b. The decomposition rate and ratio of $CF_3SO_2F$ increase with rising pressure. The decomposition ratio of $CF_3SO_2F$ reaches to 0.87 and 0.94 at 0.3 and 0.5 MPa, respectively. This trend can be attributed to the increased pressure raising the concentration of molecules, thereby increasing the probability of intermolecular collisions and accelerating the decomposition rate and ratio of $CF_3SO_2F$.

To uncover the effect of gas mixing ratios on decomposition, the time evolution of $CF_3SO_2F$ decomposition for different $CF_3SO_2F/CO_2$ gas mixture systems at 2200 K is illustrated in Fig. 2c. Compared to 12% $CF_3SO_2F/88\%CO_2$ (0.77), the decomposition ratio of 14%$CF_3SO_2F/86\%$ $CO_2$ (0.74) and 20%$CF_3SO_2F/80\%CO_2$ (0.64) within 1000 ps decrease sequentially. To address the potential influence of varying total system size, we employed two distinct modeling strategies. One strategy maintained a constant total number of molecules for all mixture ratios to directly control for system size effects. Conversely, the other strategy kept the number of $CF_3SO_2F$ reactant molecules constant and varied only the number of buffer gas ($N_2$ or $CO_2$) molecules. The decomposition results from both approaches, analyzed over consistent time spans, are presented in Fig. S4. A similar result that the increased $CF_3SO_2F$ mixing ratio would suppress the decomposition of the mixture can be observed. This may be primarily attributed to changes in molecular interactions and reaction kinetics. As the mixing ratio of $CF_3SO_2F$ increases, the intermolecular forces, such as van der Waals forces and dipole-dipole interactions between $CF_3SO_2F$ molecules[42,43], become more significant. These interactions enhance the overall stability of the system, effectively suppressing the decomposition

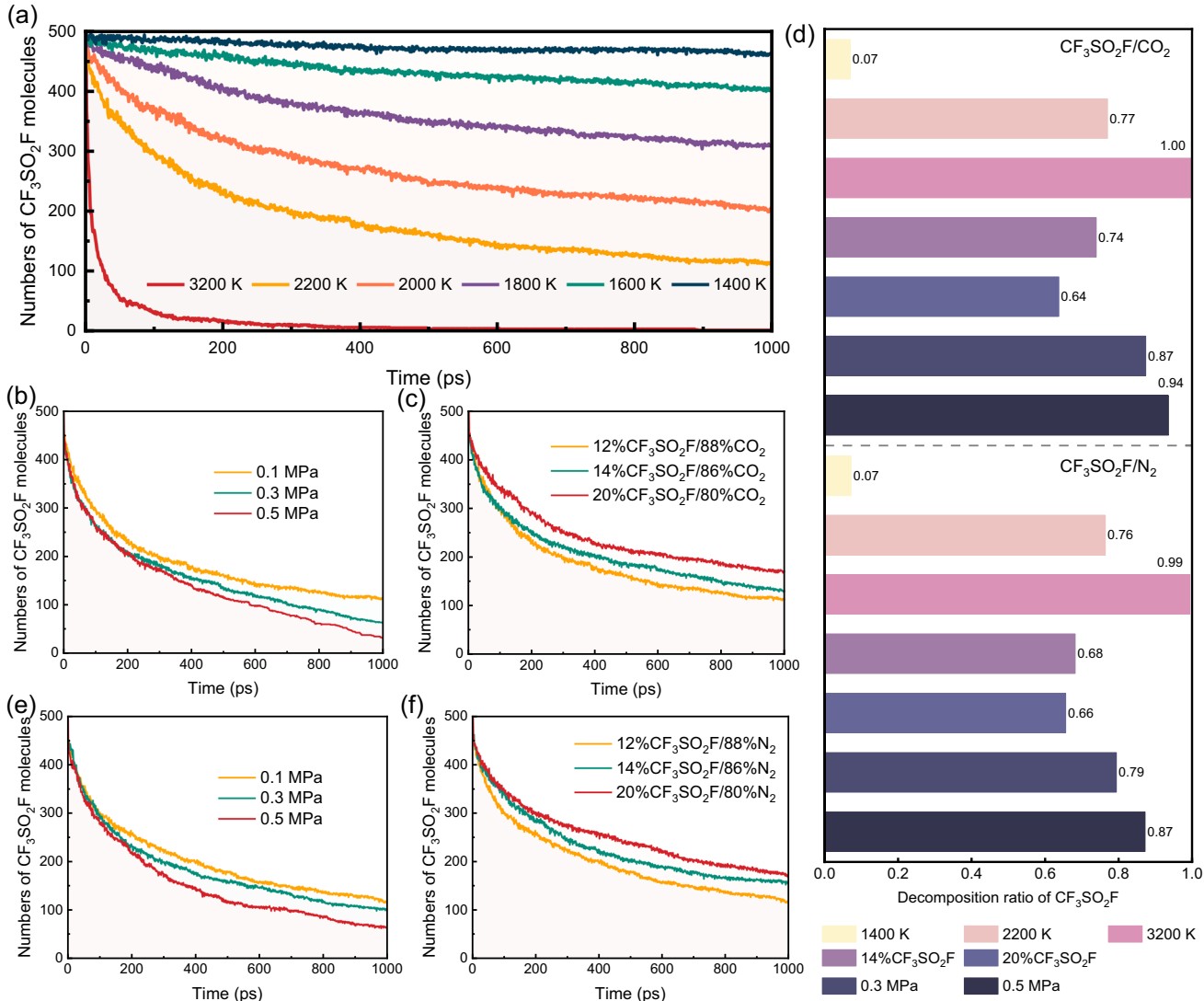

**Fig. 2 | Thermal decomposition of CF₃SO₂F under various conditions.**
**a** Decomposition time evolution at 0.1 MPa with different temperatures in 12% CF₃SO₂F/88%CO₂ mixture. **b** Decomposition time evolution at 2200 K with different pressures in 12%CF₃SO₂F/88%CO₂ mixture. **c** Decomposition time evolution at 2200 K and 0.1 MPa with different mixing ratios in CF₃SO₂F/CO₂ mixtures.

**d** Final decomposition ratios after 1000 ps under different conditions in CF₃SO₂F/ CO₂ and CF₃SO₂F/N₂ mixtures. **e** Decomposition time evolution at 2200 K with different pressures in 12%CF₃SO₂F/88%N₂ mixture. **f** Decomposition time evolution at 2200 K and 0.1 MPa with different mixing ratios in CF₃SO₂F/N₂ mixtures.

pathways. Moreover, in mixtures with a higher mixing ratio of CF₃SO₂F, a greater proportion of the collision energy is transferred into vibrational modes. Since CF₃SO₂F possesses more vibrational degrees of freedom than CO₂ as shown in Tables S2 and S3, a higher proportion of the total energy is stored in these modes rather than in translation and rotation. This reduces the fraction of effective collisions where the energy in the translational and rotational degrees of freedom exceeds the activation barrier for decomposition, thereby lowering the reaction rate. The decomposition behavior of CF₃SO₂F remained consistent across different mixture ratios, regardless of the mixing gas model employed. The consistent decomposition behavior obtained from both the constant total number and constant CF₃SO₂F number approaches confirms the robustness of our findings.

As another commonly used buffering gas, N₂ was systematically investigated to elucidate its influence on the thermal decomposition characteristics of CF₃SO₂F. Figures S5 and Fig. 2e, f present the decomposition trends of CF₃SO₂F/N₂ mixtures under various temperatures, pressures, and mixing ratios conditions. Notably, the parametric dependencies observed in N₂-based mixtures mirror those in CF₃SO₂F/CO₂ systems, confirming the generalizability of our previous findings. However, the decomposition ratios of CF₃SO₂F in N₂-buffered mixtures (ranging from 0.79 to 0.87) are notably

lower than those observed in CO₂-buffered systems (ranging from 0.87 to 0.94) at elevated pressures (0.3–0.5 MPa), as shown in Fig. 2d. This consistency was further verified by three separate simulations employing N₂ or CO₂ as buffer gases, respectively, as shown in Fig. S6. This statistically significant discrepancy indicates that N₂ offers greater effectiveness in suppressing the thermal decomposition of CF₃SO₂F, highlighting its potential as a more efficient buffering gas in high-pressure applications. The differing effects of N₂ and CO₂ buffer gases on CF₃SO₂F decomposition stem from their distinct vibrational energy transfer efficiencies. Analysis of their vibrational spectra reveals that CO₂, as a linear triatomic molecule, possesses low-frequency modes that effectively couple with the vibrational modes of excited CF₃SO₂F as shown in Tables S2 and S3. This facilitates resonant energy transfer during collisions, promoting greater energy accumulation in CF₃SO₂F and resulting in the observed higher decomposition yield in CO₂ mixtures compared to N₂, where such efficient vibrational coupling is absent.

**Main decomposition products distribution and evolution**
Figure S7a illustrates decomposition products of the CF₃SO₂F/CO₂ mixture at temperatures ranging from 1400 to 3200 K. In this study, decomposition

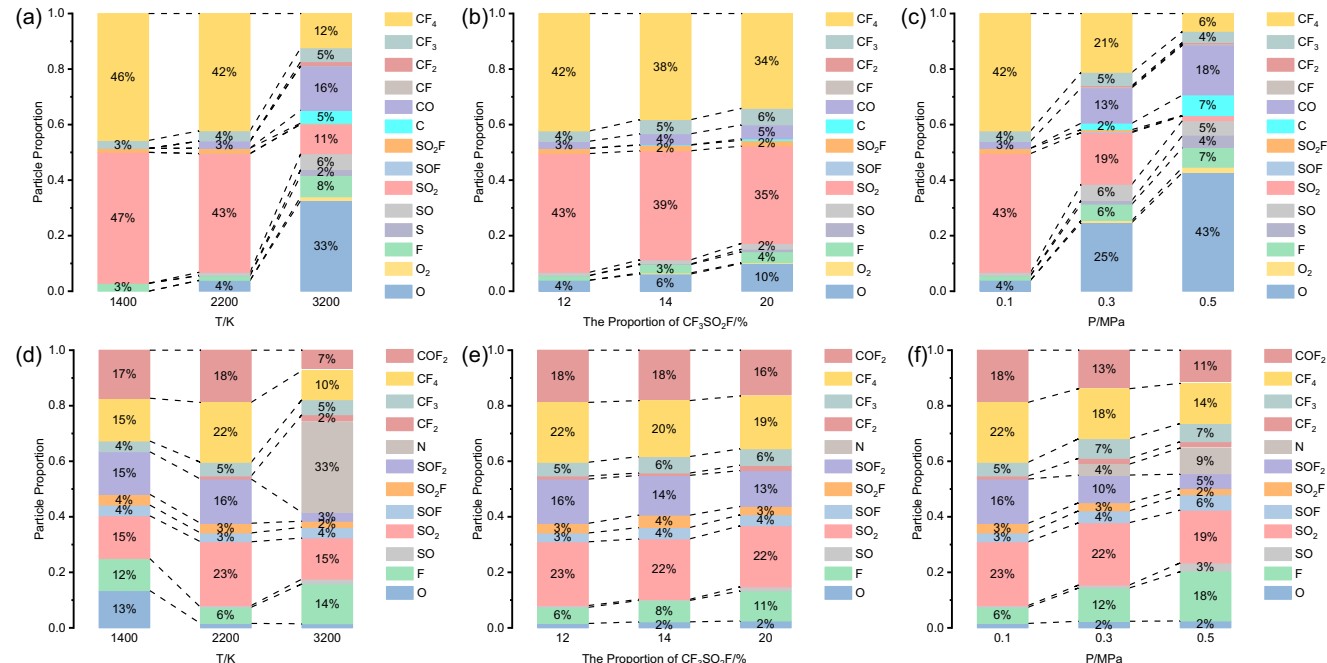

**Fig. 3 | Comparative analysis of final decomposition products in $CF_3SO_2F/CO_2$ and $CF_3SO_2F/N_2$ mixtures.** The relative concentration of $CF_3SO_2F/CO_2$ final decomposition products **a** at 1400–3200 K, 12%$CF_3SO_2F$ and 0.1 MPa, **b** at 2200 K, 12%$CF_3SO_2F$–20%$CF_3SO_2F$ and 0.1 MPa, **c** at 2200 K, 12%$CF_3SO_2F$ and 0.1–0.5 MPa. The relative concentration of $CF_3SO_2F/N_2$ final decomposition products **d** at 1400–3200 K, 12%$CF_3SO_2F$ and 0.1 MPa, **e** at 2200 K, 12%$CF_3SO_2F$–20% $CF_3SO_2F$ and 0.1 MPa, **f** at 2200 K, 12%$CF_3SO_2F$ and 0.1–0.5 MPa.

products are categorized as primary or secondary based on their formation mechanisms. Primary products, such as $CF_4$ and $SO_2$, are directly formed from the initial breakdown of $CF_3SO_2F$. Secondary products, such as SO and $CF_2$, result from subsequent reactions involving intermediate species. In the analysis of decomposition products, $CO_2$ originating from the buffer gas was systematically excluded to ensure that the reported product yields exclusively reflect new species formed from the decomposition of $CF_3SO_2F$ and its subsequent reactions. The decomposition of the $CF_3SO_2F/CO_2$ mixture begins at 1400 K. At this temperature, the extent of decomposition is limited, and the dominant reaction pathway yields $CF_4$ and $SO_2$. As the temperature increases to 2200 K, the number of primary decomposition products ($CF_4$ and $SO_2$) increase (Fig. S7a), while their relative concentrations decrease due to further decomposition into secondary products such as SO and $CF_3$ (Fig. 3a).

As the temperature increases to 3200 K, the number of chemical reactions and byproducts in the system significantly increases. At this stage, the mixture transformed into $CF_4$, $CF_3$, $CF_2$, CF, CO, C, $SO_2F$, SOF, $SO_2$, SO, S, F, $O_2$, and O. This is attributed to the further breakdown of large molecular groups into smaller molecules. For instance, $CF_4$ could decompose into $CF_3$, $CF_2$, CF, and C, while $SO_2$ may decompose into SO and S. Consequently, O atoms become the most abundant molecular fragments in the system, originating from the decomposition of both the main insulating gas $CF_3SO_2F$ and the buffer gas $CO_2$.

The formation of C and S at high temperatures indicates that solid precipitation of these elements should be considered in the design of equipment insulation. The radial distribution function (RDF) provides quantitative insight into the spatial correlation between particles within the system. Figure S8b and c present RDF analyses of the $CF_3SO_2F/CO_2$ mixture at 2200 and 3200 K, respectively. A pronounced weakening of the characteristic peaks corresponding to S–F, C–O, and S–O bonds is observed compared to the initial decomposition stage at 1400 K (Fig. S8a). This reduction directly reflects the progressive decomposition of $CF_3SO_2F$, $CO_2$, and $SO_2$ molecular species as temperature increases.

Figure S7d presents the final decomposition products of the $CF_3SO_2F/N_2$ mixture across a temperature range of 1400–3200 K. In addition to $CF_4$

and $SO_2$, the primary decomposition products consist of $COF_2$, $SOF_2$ at lower temperatures which are absent in the $CF_3SO_2F/CO_2$ mixture, suggesting distinct decomposition pathways between the two systems. Both the quantity and relative concentration of these major primary products increase as the temperature rises to 2200 K (Fig. 3d). The reaction complexity intensifies with a further increase in temperature (3200 K). $N \equiv N$ triple bond dissociates, yielding atomic N, meanwhile the primary products ($CF_4$, $SO_2$, $COF_2$, and $SOF_2$) further decompose into secondary species such as $CF_3$, $CF_2$, $SO_2F$, SOF, SO, N, F, and O. Consequently, the quantity and relative concentration of primary products decline as secondary products become dominant. Unlike the $CF_3SO_2F/CO_2$ mixture, negligible formation of oxygen and solid residues (C and S) is observed, further demonstrating the mechanistic differences between the two mixture systems. From a thermodynamic perspective, the extreme temperatures in the simulation substantially increase the entropy of atomic and radical species. Although molecular nitrogen possesses an exceptionally strong $N \equiv N$ bond, its dissociation into monatomic nitrogen becomes feasible under these conditions. The resulting free atoms are stabilized by the high-entropy environment, making the atomic state more favorable than recombination into less stable molecular products like nitrogen monoxide. Consequently, the buffer gas acts primarily through physical influences such as modulating collision frequency and energy distribution rather than as a key reactant in the core decomposition mechanism.

A comparison of the bond length distributions at 1400 K (Fig. S8d) with those at higher temperatures (Fig. S8e and f) reveals a clear temperature-dependent dissociation behavior, characterized by a broadening of the distribution and the emergence of a peak at longer distances corresponding to bond rupture. The C–S and S–F bonds, mainly associated with $CF_3SO_2F$ and $SO_2F$, the C–F bond, related to both $CF_3SO_2F$ and $CF_4$, and the S–O bond, primarily from $SO_2F$, all exhibit noticeable weakening. This indicates the progressive decomposition of these molecular species.

To further investigate the effect of $CF_3SO_2F$ mixing ratio on the decomposition products of $CF_3SO_2F/CO_2$ and $CF_3SO_2F/N_2$ mixtures, the final products quantity and the relative concentration of $CF_3SO_2F$ mixtures with different $CF_3SO_2F$ mixing ratios at 2200 K were simulated as shown in

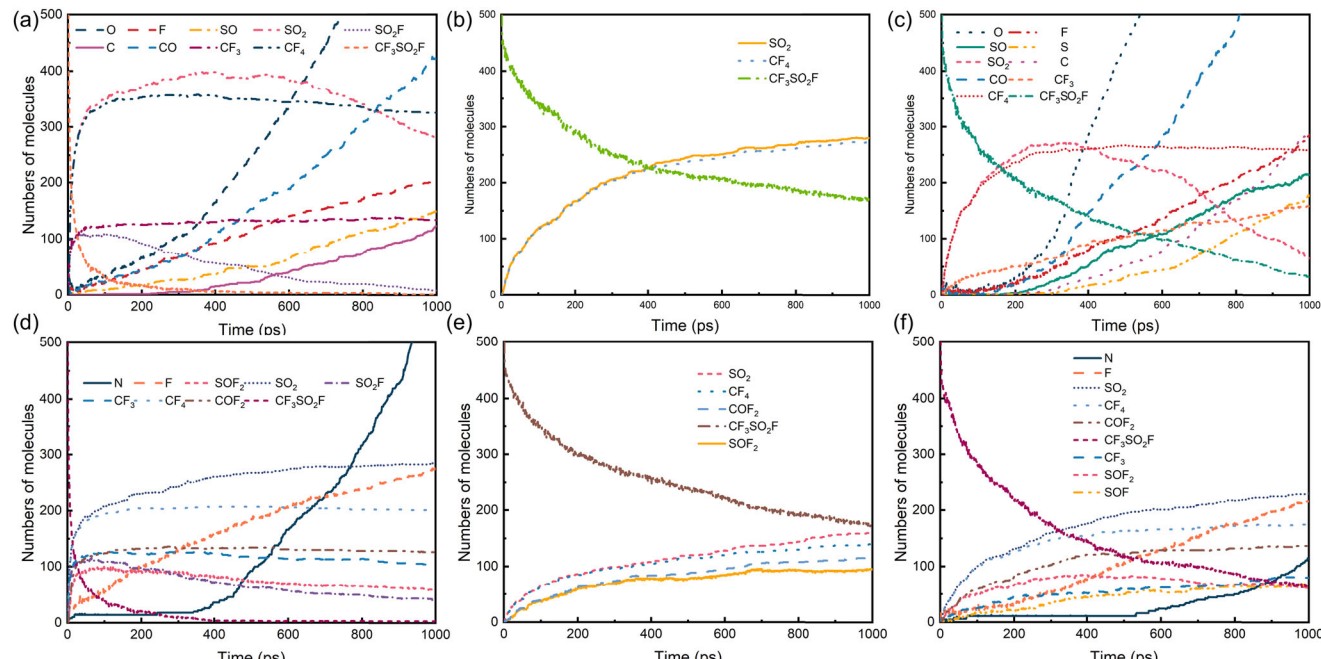

**Fig. 4 | Decomposition product dynamics in CF₃SO₂F mixtures.** Time evolution of major decomposition products in $CF_3SO_2F/CO_2$ mixtures under different conditions: **a** at 3200 K, 12%$CF_3SO_2F$ and 0.1 MPa, **b** at 2200 K, 20%$CF_3SO_2F$ and 0.1 MPa, **c** at 2200 K, 12%$CF_3SO_2F$ and 0.5 MPa. Time evolution of major decomposition products in $CF_3SO_2F/N_2$ mixtures under different conditions: **d** at 3200 K, 12%$CF_3SO_2F$ and 0.1 MPa, **e** at 2200 K, 20%$CF_3SO_2F$ and 0.1 MPa, **f** at 2200 K, 12%$CF_3SO_2F$ and 0.5 MPa.

Figures S7b, e, and 3b, e. The number of decomposition products decreases as the $CF_3SO_2F$ mixing ratio in the system increases, consistent with the $CF_3SO_2F$ decomposition evolution curves (*vs.* time) at different mixing ratios in Fig. 2c and f. Using $CF_3SO_2F/CO_2$ mixtures as an example, the $SO_2$ yield exhibits an inverse relationship with the $CF_3SO_2F$ mixing ratio. Specifically, the 12%$CF_3SO_2F$/88%$CO_2$ mixture generates 358 $SO_2$ molecules after 1000 ps under 2200 K, whereas the 20%$CF_3SO_2F$/80%$CO_2$ system produces only 280 $SO_2$ molecules. Moreover, as the mixing ratio of $CF_3SO_2F$ in the system increases (Fig. 3b), the relative concentration of $CF_4$ and $SO_2$ decreases, while the relative concentration of secondary products increases slightly. This trend aligns with the evolution of the final products quantity and relative concentration for mixture systems maintaining constant total number of molecules (Fig. S9), demonstrating the robustness of the results.

To examine the pressure dependence of $CF_3SO_2F$ decomposition products, simulations were conducted for $CF_3SO_2F/CO_2$ and $CF_3SO_2F/N_2$ mixtures at 2200 K across a pressure range of 0.1–0.5 MPa. Figures S7c and 3c illustrate the evolution of final products and product relative concentration in the $CF_3SO_2F/CO_2$ system. It can be seen that elevated pressure markedly accelerates reactions, promoting the transformation of primary products ($CF_4$ and $SO_2$) into smaller fragments, including SO, $CF_3$, and monoatomic species (C, S, F, and O). At 0.5 MPa, $CO_2$ undergoes substantial dissociation, yielding CO and O. The resulting [CO]/[$CO_2$] ratio of 0.27 in our simulations shows remarkable agreement with the value of 0.26 reported in previous studies[24]. Furthermore, the O atom is the most abundant single product (1714 particles, 43% of the total). A parallel trend is observed for $CF_3SO_2F/N_2$ (Figs. 3f, S7f), where high pressure similarly drives further fragmentation of primary products, though with comparatively lower decomposition yields.

## Decomposition mechanism of CF₃SO₂F
To elucidate the thermal decomposition mechanisms of $CF_3SO_2F/CO_2$ mixtures under varying conditions, we examined the temporal evolution of decomposition products. Both product yields and reaction rates showed strong positive correlations with temperature. Notably, the reaction profiles at 3200 K (Fig. 4a) were significantly steeper than those at 2200 K (Fig. S10),

indicating an acceleration of the reaction rate at elevated temperatures. Notably, the variation of $SO_2$ number reveals distinct reaction process: (1) an initial rapid accumulation phase (0–100 ps) characterized by a steep number increase, (2) an intermediate transition phase (100–500 ps) showing progressively slower accumulation, and (3) a final depletion phase (> 500 ps) exhibiting accelerated number decline. This behavior reflects rapid $CF_3SO_2F$ decomposition initially ($CF_3SO_2F \rightarrow CF_4 + SO_2$), producing $SO_2$ faster than its decomposition. As $CF_3SO_2F$ depletes, the formation rate of $SO_2$ slows. However, the concurrent increase in the concentrations of $SO_2$ molecules promotes its subsequent decomposition reactions, leading to its net consumption under these conditions. The presence of the product $SO_2F$ suggests that a high-temperature decomposition pathway involving direct C–S bond cleavage through intense molecular vibrations, yielding $CF_3$ and $SO_2F$ for subsequent decomposition.

The $CF_3SO_2F$ mixing ratio significantly influenced decomposition dynamics (Fig. 4b). Increasing the $CF_3SO_2F$ mixing ratio in the mixture elevated the gas density (Table 1) and suppressed $CF_3SO_2F$ decomposition, leading to a decrease in the yields of primary products ($CF_4$ and $SO_2$). Both reaction rate and product species exhibited a positive pressure dependence (Fig. 4c). At higher pressures, in addition to the primary products $CF_4$ and $SO_2$, small molecular fragments such as $CF_3$, SO, F, O, S, and C are generated in significantly greater quantities, which suggests that higher pressures promote more extensive molecular dissociation of $CF_3SO_2F$.

The decomposition of $CF_3SO_2F$ is the dominant factor governing the primary reaction pathways and product distribution in $CF_3SO_2F/CO_2$ mixtures, as shown in Fig. 5a and Table S4. Trajectory simulations reveal that the net reaction $CF_3SO_2F \rightarrow CF_4 + SO_2$ proceeds via two temperature-dependent pathways rather than a single concerted step. The primary product formation ($CF_4 + SO_2$) occurs through a concerted mechanism where C–S bond cleavage and fluorine transfer from $SO_2F$ to $CF_3$ proceed simultaneously at temperatures above 1400 K as shown in Fig. S11. This concerted process, mediated by a transition state, directly yields $CF_4$ and $SO_2$, which is consistent with previous studies[15]. At temperatures exceeding 2200 K, direct C–S bond rupture becomes increasingly prevalent, producing $CF_3$ and $SO_2F$ radicals. $SO_2F$ then decomposes primarily through S–F bond

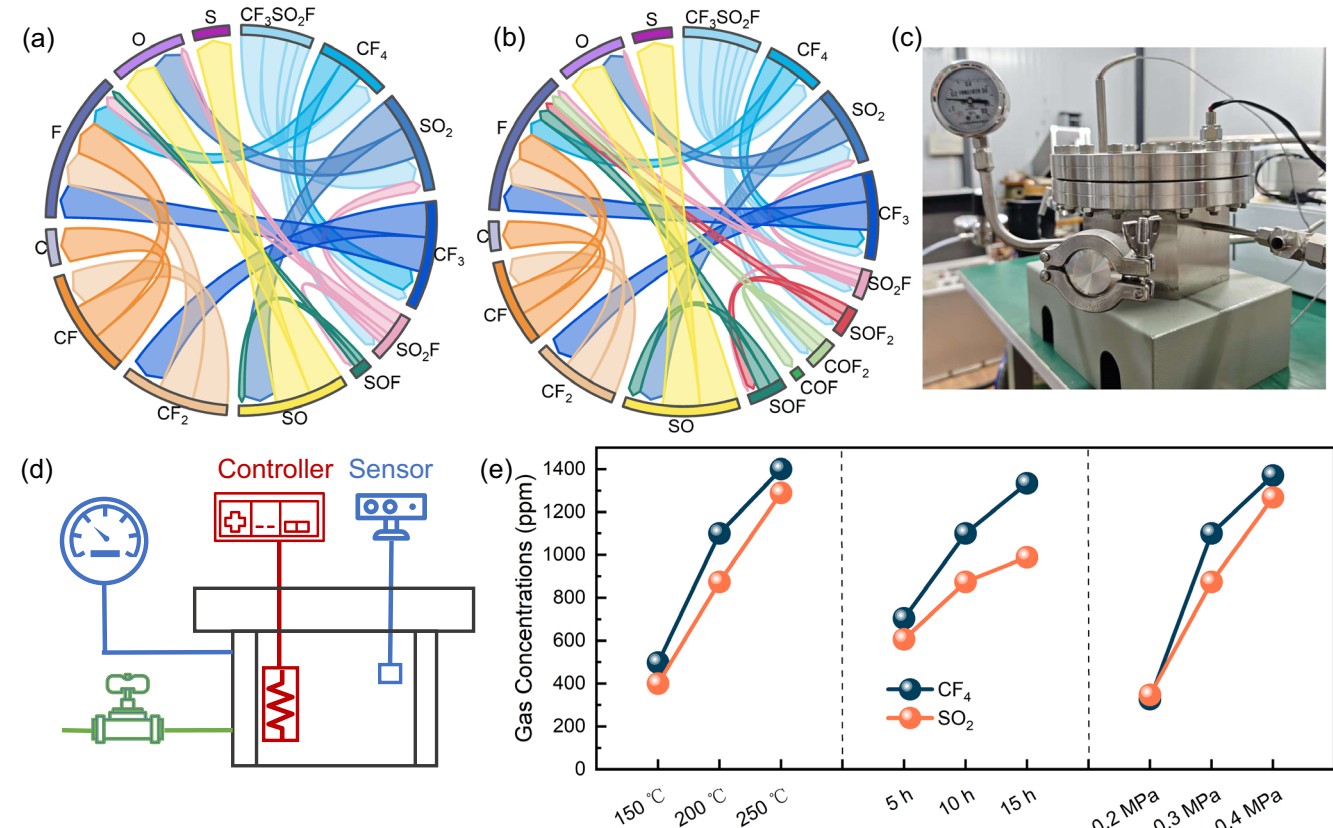

**Fig. 5 | Decomposition mechanisms and experimental validation.** Decomposition mechanisms of $CF_3SO_2F$ (**a**) in the $CF_3SO_2F/CO_2$ mixed gas and (**b**) in the $CF_3SO_2F/N_2$ mixed gas. **c** The physical setup and **d** schematic diagrams of the thermal decomposition test platform. **e** Measured gas concentrations of $14\%CF_3SO_2F/86\%N_2$ thermal decomposition under different conditions.

cleavage (yielding $SO_2$ + F) with minor S–O cleavage (producing SOF + O). The resulting SOF decomposes further to SO and F. Primary products undergo progressive defluorination and deoxygenation, ultimately generating atomic species.

The $CF_3SO_2F/N_2$ system demonstrates decomposition mechanisms analogous to $CF_3SO_2F/CO_2$ while displaying distinct kinetic and product characteristics. At lower temperatures, $CF_3SO_2F$ decomposition with $N_2$ existing proceeds predominantly through two competing pathways: (1) direct formation of primary product of $SO_2$ and $CF_4$ ($CF_3SO_2F \rightarrow CF_4 + SO_2$), and (2) direct formation of secondary product of $COF_2$ and $SOF_2$ ($CF_3SO_2F \rightarrow COF_2 + SOF_2$) as shown in Fig. S12. Upon increasing the temperature to 3200 K, an additional pathway emerges involving direct C–S bond cleavage to yield $CF_3$ and $SO_2F$ radicals (Fig. 4d), mirroring the behavior observed in $CO_2$-based mixtures. Notably, in the $N_2$ buffer gas, variations in the $CF_3SO_2F$ mixing ratio show minimal impact on the fundamental reaction pathways, as evidenced by the consistent final product species across different mixture ratios (Figs. 4e and S12), even though the absolute product yields vary with the initial reactant amount. However, the system exhibits significantly reduced product diversity and slower reaction kinetics under elevated pressures (Fig. 4f), confirming the superior efficacy of $N_2$ as a decomposition inhibitor compared to $CO_2$. The distinct decomposition mechanism of $CF_3SO_2F/N_2$ (Figs. 5b and S13) primarily differs through the inclusion of the $COF_2/SOF_2$ formation pathway. The absence of the $COF_2/SOF_2$ formation pathway in the $CO_2$ system contributes to the observed differences in product profiles and decomposition rates between the two mixtures.

### Experimental validation of $CF_3SO_2F$ decomposition

To validate the simulation results, an overheating decomposition experiment was performed to investigate the thermal decomposition characteristics of $14\%CF_3SO_2F/86\%N_2$ gas mixtures. The experiments were

systematically conducted under varying temperatures (150–250 °C in 50 °C increments), overheating duration (5h–15 h in 5 h increments), and gas pressures (0.2–0.4 MPa in 0.1 MPa increments), corresponding to the typical operational range of high voltage gas insulated equipment.

Following each test, the residual gas composition in the reaction chamber was sampled and analyzed. The GC-MS test result of the decomposition products of $14\%CF_3SO_2F/86\%N_2$ at 200 °C for 15 h is illustrated in Fig. S14. $CF_4$ and $SO_2$ were identified as the dominant decomposition products. Consequently, their yield variations under different conditions were systematically analyzed. As shown in Fig. 5e, the quantitative results demonstrate that $CF_4$ and $SO_2$ are the primary decomposition products with comparable concentrations, which aligns with the key predictions of the computational model. Furthermore, the observed consistent increase in their yields with rising temperature, extended reaction time, and elevated pressure firmly establishes the validity of the proposed decomposition mechanism. Furthermore, the radicals generated during decomposition exhibit a low tendency to recombine into $CF_3SO_2F$, further confirming the irreversible decomposition process.

### Discussion

In summary, the decomposition mechanism of $CF_3SO_2F/CO_2$ and $CF_3SO_2F/N_2$ gas mixture at different temperatures, gas mixing ratios, and pressures was theoretically investigated and experimentally verified by deep learning potential training and molecular dynamics simulation. The results show that the decomposition of $CF_3SO_2F/CO_2$ gas mixture starts at 1400 K under the time scale of 1000 ps, and the primary decomposition products are $CF_4$ and $SO_2$. When the temperature increases to 3200 K, $CF_3SO_2F$ will decompose rapidly in the arc discharge channel, and the decomposition ratio of $CF_3SO_2F$ reaches 0.94 within 1000 ps and decomposes to $CF_4$, $CF_3$, $CF_2$, CF, CO, C, $SO_2F$, SOF, $SO_2$, SO, S, F, $O_2$, O. At lower temperatures, the $CF_3SO_2F$ molecule tends to cleave the C–S bond, with the F atom on $SO_2F$

transferring to $CF_3$ during the bond breakage, ultimately forming $CF_4$ and $SO_2$. In contrast, at higher temperatures, increased molecular vibrations cause the C–S bond to break more directly, leading to the formation of $CF_3$ and $SO_2F$, which then undergo further decomposition. Furthermore, a higher initial concentration of $CF_3SO_2F$ increases the absolute number of decomposition events. The accelerated accumulation of primary products such as $CF_4$ and $SO_2$ favors their own secondary reactions. The initial decomposition rate of $CF_3SO_2F$ is significantly promoted by elevated system pressure due to the corresponding increase in molecular collision frequency. For engineering applications, this finding highlights the essential need for rigorous condition monitoring of electrical equipment to prevent significant $CF_3SO_2F$ decomposition due to localized overheating, which would degrade its electrical insulation performance.

The primary decomposition pathways of $CF_3SO_2F/N_2$ and $CF_3SO_2F/CO_2$ mixtures are similar, except for an additional reaction in the latter system ($CF_3SO_2F \rightarrow COF_2 + SOF_2$). The effects of temperature, gas composition, and pressure on the decomposition behavior of $CF_3SO_2F$ in $N_2$ closely resemble those observed in $CO_2$. However, under high-pressure conditions, $N_2$ acts as an inert buffer gas and more effectively suppresses the decomposition of $CF_3SO_2F$ compared to $CO_2$.

While $CF_3SO_2F$ is proposed as an environmentally friendly alternative to $SF_6$, the potential impact of its decomposition products should be rigorously assessed. Our simulations identify $CF_4$ and $COF_2$ as key decomposition species. It is critical to note that while the GWP of $CF_4$ (7390) is significantly lower than that of $SF_6$ (23500), it remains higher than that of carbon dioxide, with an atmospheric lifetime exceeding 50,000 years. Furthermore, $COF_2$ is a highly toxic compound, posing potential safety hazards in the event of insulator failure and gas release. Our results, however, provide crucial insights for mitigating these risks. The formation of both $CF_4$ and $COF_2$ exhibits a strong dependence on temperature and the buffer gas environment. Specifically, using $N_2$ as a buffer gas can suppress the formation of these harmful byproducts, potentially serving as an effective strategy to minimize environmental and safety risks. Therefore, the viability of $CF_3SO_2F$ as an $SF_6$ replacement hinges not only on its intrinsic properties but also on engineering controls that optimize operating conditions to limit the formation of deleterious by-products. This molecular-level understanding directly informs the development of safer, next-generation gas-insulated equipment aligned with global decarbonization goals.

In conclusion, this study identifies the major decomposition products and elucidates the primary reaction mechanisms of $CF_3SO_2F$ under high-temperature and high-pressure conditions. The established computational framework provides a foundation that can inform future assessments of environmental impact and guide the development of safer $CF_3SO_2F$-based insulation technologies.

## Methods
### AIMD calculations setup
AIMD simulations were executed to construct the training dataset for machine learning. All calculations were carried out in the Vienna ab initio simulation packages (VASP)[44,45]. Ion-electron interactions were modeled with the Projector Augmented Wave (PAW) method[46]. Using the revised Perdew−Burke−Ernzerhof functional for describing electronic exchange and correlations, DFT calculations were executed employing the generalized gradient approximation method[47]. The plane-wave basis was applied with a cutoff energy of 520 eV. The convergence criteria for the electronic energy and structural relaxation were set to $10^{-6}$ eV and 0.01 eV/Å, respectively. AIMD simulations were performed in the canonical ensemble (NVT) with periodic boundary conditions and a time step of 0.5 fs to obtain configurations, energies, forces, and virials over a wide range of temperatures for the $CF_3SO_2F$ gas mixture with three $CF_3SO_2F$ molecules and six buffer gas molecules in the box. In the NVT simulations, the Nosé-Hoover thermostat was employed to maintain isothermal conditions at 30–3200 K for 20 ps. Molecular species were identified from the MD trajectories using a topology analysis algorithm based on interatomic distances and bond orders. This method allows for the dynamic tracking of bond formation and dissociation, enabling continuous identification of all chemical species throughout the simulation. The analysis script is available in the associated GitHub repository. Details of DP-GEN, DLP training, and MD setup were listed in the Supplementary Methods section of the Supporting Information.

### Thermal decomposition characteristics test platform
The experimental platform for investigating thermal decomposition characteristics comprises three main components: (1) a high-temperature test chamber, (2) a temperature control system, and (3) a DC power supply. The stainless steel test chamber (340 L grade, 10 L capacity) operates within a pressure range of 0–0.4 MPa. The thermal decomposition tests were conducted using a custom experimental platform, with the physical setup and schematic diagrams shown in Fig. 5c, d. The system pressure was monitored by a high-precision digital barometer connected to the chamber. A temperature control system, integrating electromagnetic relays with sensors and controllers, maintained precise thermal conditions. A DC power supply energized thermocouples to simulate the localized overheating faults typical in gas-insulated equipment. The gas composition and relative concentration of $CF_3SO_2F$ mixed gas after the test were analyzed by gas chromatography–mass spectrometry (GC–MS) equipped with a GS-GASPRO column (60 m). The inlet temperature was maintained at 100 °C with a split ratio of 20:1. Helium carrier gas was used at a constant flow mode with a total flow of 57.0 mL/min, a column flow of 2.43 mL/min, and a linear velocity of 39.7 cm/s. The oven temperature program was as follows: held at 40 °C for 1 min, ramped to 120 °C at 7 °C/min, and finally held at 120 °C for 6 min. The MS ion source and transfer line temperatures were both set to 200 °C. Species identification was achieved by matching the acquired mass spectra and retention times against reference standards.

## Data availability
All data supporting the findings of this study are available within the article and its Supplementary Information. Source data are available as Supplementary Data 1.

## Code availability
The DLP models are available on GitHub at https://github.com/LZYUCL/DLP_CF3SO2F and archived on Zenodo at https://doi.org/10.5281/zenodo.17731042.

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

## Acknowledgements
This work was supported by the National Key Research and Development Program of China (2021YFB2401400).

## Author contributions

Anyang Wang, Zeyuan Li, and Shubo Ren contributed equally to this work. Anyang Wang: writing—original draft, investigation, visualization. Zeyuan Li: validation, visualization, software. Shubo Ren: methodology, investigation, validation. Xue Ke: investigation, visualization. Xuhao Wan: validation, visualization. Rong Han: investigation. Xianglian Yan: resources, Wen Wang: investigation. Yu Zheng: resources, methodology, supervision. Yuzheng Guo: writing—review and editing, supervision. Jun Wang: writing—review and editing, supervision, funding acquisition, conceptualization.

## Competing interests

The authors declare no competing interests.
