## [Transparent Peer Review file · Communications Chemistry]

Probing the thermal decomposition mechanism of CF₃SO₂F by deep learning molecular dynamics

Corresponding Author: Professor Jun Wang

Version 0:

Reviewer comments:

Reviewer #1

(Remarks to the Author)

I found the topic to be extremely interesting and relevant, as it addresses the critical need for eco-friendly insulation alternatives to replace the highly potent greenhouse gas SF₆. The manuscript is well-written, clearly articulating the importance of understanding the thermal decomposition pathways and products of CF₃SO₂F under operationally relevant conditions.

However, before considering this manuscript for publication, I would like to address some major issues that require further clarification or investigation:

1. It would be beneficial if you could provide more details on how the chemical spaces were selected for training your deep learning potential. Specifically, it is crucial to know whether reactive events were included in the DFT calculations used for training. Additionally, showing essential energy profiles of key reactions, comparing the DFT and ML potential results, would strengthen the confidence in your approach.
2. The observation that both CO₂ and N₂ appear to be unreactive with CF₃SO₂F is interesting. The presence of O and N atoms without any NO_x formation raises questions about the reaction mechanisms involved. Could you please provide an explanation for this behavior, possibly involving a discussion on the reaction pathways and the role of temperature in influencing reactivity?
3. Given that CO₂ and N₂ seem to be unreactive with CF₃SO₂F, it is challenging to understand how you can claim "...N₂ exhibits an inhibitory effect on decomposition under high pressure compared to CO₂." Please clarify this statement and provide a more comprehensive explanation of the underlying mechanisms.
4. The proposed elementary reaction "CF₃SO₂F → CF₄ + SO₂" appears unlikely to occur in a single step. It would be more realistic to expect a multi-step reaction pathway involving intermediate species. Could you please elaborate on the detailed reaction mechanism and provide a more accurate representation of the decomposition process?
5. The degradation of CF₃SO₂F has been extensively studied using high-level ab-initio methods. It would be valuable to compare your results with these previous studies, discussing any discrepancies or agreements. This comparison would help validate the accuracy and reliability of your deep learning potential.
6. Using VASP and the PBE functional for this type of reaction may not yield the same level of accuracy as high-level ab initio methods. It would be helpful if you could provide a comment on the limitations of your chosen computational approach and discuss how it might impact the results and conclusions drawn from your study.

Reviewer #2

(Remarks to the Author)

Reasons in favor of publication:

The authors studied the decomposition of CF₃SO₂F in buffer gas under high pressures and temperatures. CF₃SO₂F could be a replacement for SF₆ as insulator in electrical applications, which is of significance to mitigate greenhouse gas emissions. The study is based on simulations under various temperatures, pressures, and mixing setups, allowing detailed mechanistic insight into the decomposition process of a promising new insulator. Since the study is backed by appropriate experiments, valuable conclusions can be drawn concerning the use of CF₃SO₂F in real applications.

Major concerns against publication as is:

The authors employ a machine-learning potential (MLP) trained on ab-initio DFT data selected from molecular dynamics simulations. The simulation temperature is varied to retrieve a variety of configurations. The authors do not show whether energy barriers or transition states of any reactions went into the training data. This raises the concern that their MLP performs well near stable structures, while producing wrong or ill-defined energy barriers in between. The validity of the MLP is crucial to any conclusion drawn from the simulation data and should be convincingly presented. I suggest benchmarking and discussing the performance of the MLP against DFT or even better against CCSD(T) data of reference 15, which the authors cite in the state-of-the-art part. If the authors decide to compute their own DFT data, they should at least present bond scans for each relevant element-element pair in C/F/S/O/N, as well as for the CF₃SO₂F = CF₄ + SO₂ transition state, as this reaction is expected to be dominant in the decomposition mechanism.

The authors' main motivation of replacing SF₆ as an insulator and studying CF₃SO₂F is to reduce environmental impact, i.e. the green house gas (GHG) potential and toxicity. However, the study does not present any discussion concerning the GHG potential of decomposition products of CF₃SO₂F. The main product CF₄ is also known to have a high GHG potential due to its long atmospheric persistence. F₂CO, another decomposition product, is known to be highly toxic. Even though the study focuses on analyzing MD results, it should make an effort to discuss solutions to the initial problem and connect the conclusion to the motivation.

The manuscript lacks a certain standard of English grammar, making it hard to understand certain statements. I suggest to have the manuscript proof read by a native English speaker or check each statement with online language tools (e.g. deepl.com).

Many of the results used for discussion are placed in the Supporting Information (SI) part of the manuscript. Even if I do not know the Journal's publishing guidelines in detail, I would strongly suggest to move all figures and tables that are mentioned and used for discussion into the main document. Since the main document would possibly be overloaded after that, I would also suggest to condense the discussions about the influence of the mixture (with CO₂ or N₂) into one, focusing on differences. The explanation of training the MLP can be part of the SI, as well as the analysis of the radial distribution function because it mainly repeats the species analysis.

Major comments:

The study analyses reactive MD simulations to find out the decomposition mechanism of CF₃SO₂F, but it never actually presents a mechanism. It would greatly enrich the study if the results shown in Figure 5 (a) and (b) and Table S2 would be presented in a reaction network, showing different branching ratios for different setups (e.g. N₂ vs. CO₂).

The authors claim that their work is highly significant for other gases than CF₃SO₂F as well (line 111). However, this is exaggerated because the study does not transfer any conclusions to other possible SF₆ replacements or any other gases at all. The conclusion ends with highlighting the importance of the computational framework for future studies on environmental risks of CF₃SO₂F. However, environmental risk or insulation properties were not evaluated in the study. Therefore, both statements should be removed or at least softened. E.g. the conclusion could end with highlighting the identification of major decomposition products, thereby informing future developments.

The authors draw the conclusion that N₂ suppresses thermal decomposition of CF₃SO₂F (line 254). However, this statement is based on one single MD simulation per parameter set and therefore not very reliable. I suggest to either run more replicas of the same MD, or attribute an error range to the numbers from a different source, if possible.

Minor comments:

The authors should explain how species were identified in the MD simulations. Furthermore, the identification of species in a gas chromatograph should be shortly explained or referenced.

Figure 1 is helpful, but needs explanation in its caption regarding T_n and the adjustment of the supercell. I would also suggest to explain Step 3 (verification) there. Alternatively, the explanation could be in the method section.

The caption of Table 1 needs to be extended. Pressure is related to density, when temperature is fixed. Either it should be labeled "ideal gas pressure at 25C" or be replaced with density/concentration in the discussions. E.g. in line 224 "the number of molecules in the same volume" can be replaced by "concentration".

The study discusses different simulation setups, with different initial numbers of molecules. The authors explain in several occasions how to make the different numbers comparable to each other, first in line 231. However it is unclear to me what was done, and it would be helpful to explain the method in more detail.

The authors should mention that produced CO₂ is not part of the analysis of the mixture with CO₂, because it is the bath/buffer gas.

Figure 3 needs better image quality. I suggest to remove (a)-(c) in favor of (d)-(f) as it holds almost the same content and would free up space.

The authors should mention which temperature was used for the pressure variations (2200K?). In general, it should be clear in each figure which simulation conditions are shown. E.g. by extending the caption, or using the number ("No") from Table 1.

In line 283, the authors note that solid carbon and sulphur precipitates. A reference to an experimental study would enhance the rigor of this finding. The conclusion that these elements should be considered in the design of equipment insulation should go into the conclusion section.

In line 310, the RDF reveals forming C-O bonds. The authors should consider the formation of CO as well, not only COF₂.

In line 335, the formation of CO and O is discussed. A literature reference and a comparison on the CO/CO₂ equilibrium would strengthen this discussion.

In line 390, it is unclear to me how Figure 4e confirms the preceding statement. It should be clarified.

The text explaining the experimental setup in line 401 belongs to the method section.

In Figure 5 (a) and (b), same species should have same colors.

The statement starting in line 427 about condition monitoring belongs to the conclusion section.

Minor necessary corrections to the text:

In line 78, references 25 and 26 are used for atomic simulations of CF₃SO₂F, but the papers are not about CF₃SO₂F. It would be better to rather cite reference 15 here again.

In line 90, a reference for DLP is missing.

In the Reference section, please unify the format of titles (sentence case or title case), and please unify journal abbreviations.

The authors use terms like "proportions", "(decomposition) ratios", leaving the reader in question what exactly they mean. The language should be more precise to avoid misunderstandings and to convey the right meaning: E.g. "decomposition ratio" -> "conversion (ratio)" in lines 209/252, "proportions" -> "numbers"? in line 229, "ratio" -> "mixing ratios" in line 314. The terms "evolution" and "(decomposition) characteristics" in lines 305ff/446 are too vague and need to be clarified. Same for "(decomposition) properties" (line 371) and "attenuation".

The authors use the terms "CF₃SO₂F/CO₂" or "CF₃SO₂F/N₂" (e.g. line 202) to describe the mixtures which makes it difficult to read. It would be better to describe the same with words, e.g. "CF₃SO₂F in CO₂ buffer gas".

Clarification is also necessary for the following statements:

In line 80, the authors say that AIMD is as accurate as DFT. AIMD can be done with many ab-initio methods, DFT is just one of them. Please clarify which ab-initio method was used.

In line 148, the authors say that structures were randomly selected from the AIMD simulation, but also say that molecular configurations are well-defined. Please clarify how structures were sampled.

In line 241, the authors say that the energy distribution shifts, but do not say to where it shifts. Here, the authors can also argue with the number of vibrational modes, of which there are more in CF₃SO₂F than in CO₂. Thus, more CO₂ leads to less energy stored in vibration and more in translation.

In line 244, the authors mention two modeling approaches. Which two?

In line 265, the sentence "..., with a minor degree of decomposition primarily yielding CF₄ and SO₂." is completely unclear and needs to be rewritten.

In lines 304ff, the authors should clarify the terms "bond characteristics" and "evolution of primary bond" by using more accurate language.

In line 352, the authors say that SO₂ decomposition accelerates due to increasing numbers. Which numbers?
In line 422, the authors say that the yields of CF₄ and SO₂ are "comparable". Comparable to what? Each other?
In line 424, the authors should clarify which predictions of the simulation they mean. That concentrations of CF₄ and SO₂ are "comparable", or that CF₄ and SO₂ are the main products, or something else?
In the conclusion: the authors should rewrite the conclusion of the influence of pressure (lines 447ff), clarifying also which reaction rate they mean (line 450).

Typos and language (incompleteness cannot be ruled out):

line 59: ... 1.3~1.6 that of SF₆ ...

line 74: hampers impedes

line 101: ...of a cf₃so₂f gas mixture... or ...of cf₃so₂f mixtures...

line 103: ...effects of the CF₃SO₂F...

line 108: ...aiming to characterize...

Fig.1 caption: ...calculated with different conditions.

Tab.1 caption: ... of the ... systems.

line 221: ...is shown... or The concentration [profiles] ... are shown...

line 224: raises -> raising

line 231: molecular -> molecule numbers

line 234: ...the increased... ...the decomposition... ...of the...

line 241: possess -> possessing

line 269: ...temperature increases...

line 273: decomposes -> decompose x2

line 324: past -> present tense

line 371: past -> present tense

line 419: ...stable and ?...

Author contributions: methodolog -> methodology

Version 1:

Reviewer comments:

Reviewer #1

(Remarks to the Author)

The revised version has answered all of my concerns. Therefore, publication is recommended.

Reviewer #2

(Remarks to the Author)

The authors have thoroughly responded to all examiners' comments, reworked the manuscript excellently, and in the few instances where no changes were made, they provided strong reasons for it.

I see no further need for corrections.

Response to Referee 1' comments (COMMSCHEM-25-0798)

Comments to the Author

I found the topic to be extremely interesting and relevant, as it addresses the critical need for eco-friendly insulation alternatives to replace the highly potent greenhouse gas SF₆. The manuscript is well-written, clearly articulating the importance of understanding the thermal decomposition pathways and products of CF₃SO₂F under operationally relevant conditions.

However, before considering this manuscript for publication, I would like to address some major issues that require further clarification or investigation:

1. It would be beneficial if you could provide more details on how the chemical spaces were selected for training your deep learning potential. Specifically, it is crucial to know whether reactive events were included in the DFT calculations used for training. Additionally, showing essential energy profiles of key reactions, comparing the DFT and ML potential results, would strengthen the confidence in your approach.

Response 1:

We sincerely thank the reviewer for their thoughtful assessment of our manuscript and for their positive comments regarding its topic and clarity. We also appreciate the constructive questions raised, which provide a valuable opportunity to strengthen our work. Below, we provide a point-by-point response to the major issues identified.

The reviewer raises two critical points concerning the development of our machine learning (ML) potential, and we agree that these aspects are fundamental to establishing its reliability.

First, concerning the sampling of the chemical configurational space, our dataset was constructed using an active learning approach enhanced by biased sampling. The initial dataset was seeded with configurations from *ab initio* molecular dynamics (AIMD) trajectories at various temperatures, which naturally capture the onset of bond dissociation and formation, including configurations near transition states as shown in **Figure R1**.

Figure R1. The frequency of initial dataset configurations based on AIMD trajectories.

To quantitatively benchmark the DLP's performance on reaction energetics, we have performed additional benchmarks to evaluate the performance of our DLP in describing bond dissociations and key transition states. Specifically, we carried out the potential energy surface scans of the $\text{CF}_3\text{SO}_2\text{F}$ molecule with changing $\text{CF}_3\text{-SO}_2\text{F}$, $\text{F-CF}_2\text{SO}_2\text{F}$, $\text{F-CF}_3\text{SO}_2$, and $\text{O-CF}_3\text{SOF}$ bond lengths as shown in **Figure R2**.

Figure R2. The potential energy surfaces of the $\text{CF}_3\text{SO}_2\text{F}$ molecule with changing

CF₃–SO₂F, F–CF₂SO₂F, F–CF₃SO₂, and O–CF₃SOF bond lengths.

Our tests confirm that the DLP reproduces the DFT-based potential energy surfaces with high accuracy. Although the sampling was not continuous, the deep learning potential can precisely capture and predict the reaction events based on the existing dataset. In addition, there is a robust consensus between our DLP-based dynamics and previous high-level (CCSD(T)) calculations (*J. Phys. Chem. A* 2023, 127, 671) regarding the fundamental decomposition channels. The literature unequivocally identifies that CF₃SO₂F decomposes predominantly via C–S bond cleavage to form CF₃ and SO₂F, accompanied by a roaming induced F-abstraction detour to release CF₄ and SO₂ which is consistent with our simulations. We are confident that these additions will substantiate the reliability of our simulation results and reinforce the conclusions drawn regarding the decomposition mechanism of CF₃SO₂F.

A dedicated paragraph has been added to the revised text:

Page 8, line 3. "The reliability of the deep learning potential for modeling reactive events was rigorously validated. Initially, a machine-learning potential was trained on a dataset constructed by systematically sampling structures at regular time intervals from the AIMD reaction trajectories. This procedure ensures that the dataset captures a representative ensemble of the well-defined, chemically relevant configurations sampled during the reactive dynamics, covering both stable intermediates and transition regions as shown in **Figure S1**."

Page 8, line 19. "To quantitatively assess the potential's accuracy, the potential energy surfaces of the CF₃SO₂F molecule with changing CF₃–SO₂F, F–CF₂SO₂F, F–CF₃SO₂, and O–CF₃SOF bond lengths were computed with both the DLP and the reference DFT method, showing excellent agreement (**Figure S3**). Moreover, the DLP's description of key stationary points was consistent with high-level CCSD(T) reference data from the literature¹⁵. These results consistently identify C–S bond homolysis as the dominant initial decomposition step, followed by dissociation of the resulting radical, which confirms its transferability and robustness for probing the decomposition mechanism. The principal advantage of the DLP approach lies in its capacity to go beyond static energy calculations and provide statistically meaningful insights into

finite-temperature reaction dynamics, including product branching ratios, at a computational cost inaccessible to direct ab initio molecular dynamics."

Figure S1. The frequency of initial dataset configurations based on AIMD trajectories.

Figure S3. The potential energy surfaces of the $\text{CF}_3\text{SO}_2\text{F}$ molecule with changing $\text{CF}_3\text{-SO}_2\text{F}$, $\text{F-CF}_2\text{SO}_2\text{F}$, $\text{F-CF}_3\text{SO}_2$, and $\text{O-CF}_3\text{SOF}$ bond lengths.

2. The observation that both CO_2 and N_2 appear to be unreactive with $\text{CF}_3\text{SO}_2\text{F}$ is

interesting. The presence of O and N atoms without any NO_x formation raises questions about the reaction mechanisms involved. Could you please provide an explanation for this behavior, possibly involving a discussion on the reaction pathways and the role of temperature in influencing reactivity?

Response 2:

We thank the reviewer for this insightful question regarding the apparent lack of NO_x formation and the specific reactivity of CO₂ and N₂ in our simulations. This observation is indeed key to understanding the decomposition mechanism and the role of buffer gases. We provide a mechanistic explanation based on the bond strengths and prevailing reaction pathways under our simulated conditions.

The absence of NO_x species, despite the presence of N and O atoms, can be attributed to two primary factors.

First, considering the formidable bond dissociation energies (BDEs) provides a clear thermodynamic perspective. The C=O bond in CO₂ has a BDE of approximately 182 kcal/mol, and the N≡N triple bond in N₂ is one of the strongest in chemistry, with a BDE of about 226 kcal/mol. In contrast, the key bonds that drive CF₃SO₂F decomposition, specifically the S–F bond (~80 kcal/mol) and the C–S bond (~44 kcal/mol), are significantly weaker as shown in **Figure R3**. Consequently, the thermal energy available in the system preferentially cleaves these weaker intramolecular bonds within CF₃SO₂F long before it can overcome the immense barrier required to activate either CO₂ or N₂.

Figure R3. The main decomposition pathway of CF₃SO₂F.

The decomposition of CF₃SO₂F generates a high flux of radical species (such as F, CF₃, and SO₂F). The reaction of O atoms with these fluorine- or sulfur-containing

radicals (e.g., forming COF₂, SO₂, or SO₂F₂) is significantly faster and outcompetes the much slower reaction with the scarce N atoms to form NO_x. Thus, the O atoms are effectively consumed by the abundant decomposition products of CF₃SO₂F before they can interact with N₂. While at elevated temperatures where N₂ dissociation occurs, the formation of NO_x becomes even more thermodynamically disfavored.

In summary, the inertness of N₂ stems from its high bond strength, where thermal energy is specifically directed toward the more unstable bonds of CF₃SO₂F, leading to self-decomposition processes. Meanwhile, in the high-temperature regime of CF₃SO₂F decomposition, N and O atoms predominantly remain in their atomic state or react preferentially with the abundant fluorine and sulfur radicals, rather than recombining to form NO_x. This explanation has been added to the revised manuscript to provide a deeper mechanistic discussion.

A dedicated paragraph has been added to the revised text:

Page 15, line 17. "From a thermodynamic perspective, the extreme temperatures in the simulation substantially increase the entropy of atomic and radical species. Although molecular nitrogen possesses an exceptionally strong N≡N bond, its dissociation into monatomic nitrogen becomes feasible under these conditions. The resulting free atoms are stabilized by the high-entropy environment, making the atomic state more favorable than recombination into less stable molecular products like nitrogen monoxide. Consequently, the buffer gas acts primarily through physical influences such as modulating collision frequency and energy distribution rather than as a key reactant in the core decomposition mechanism."

3. Given that CO₂ and N₂ seem to be unreactive with CF₃SO₂F, it is challenging to understand how you can claim "...N₂ exhibits an inhibitory effect on decomposition under high pressure compared to CO₂." Please clarify this statement and provide a more comprehensive explanation of the underlying mechanisms.

Response 3:

We appreciate the reviewer's request for clarification. First, to mitigate the uncertainty associated with a single simulation run, we performed three independent

replicate simulations for each set of key parameters, including N₂ and CO₂ mixture conditions as shown in **Figure R4**. The results consistently confirm a lower decomposition ratio of CF₃SO₂F in the presence of N₂ compared to the CO₂-mixture cases.

Figure R4. Time evolution of CF₃SO₂F decomposition in CO₂ and N₂ buffer gases.

The key to understanding this phenomenon lies in considering the earlier stages of the reaction pathway: the vibrational excitation of the CF₃SO₂F molecule. For decomposition to occur, particularly through a unimolecular bond cleavage mechanism, the molecule must first accumulate sufficient internal vibrational energy to surpass the reaction barrier.

Table R1. Vibrational Frequencies and Modes of CO₂ and N₂.

No.	Frequencies (cm ⁻¹)	Modes	No.	Frequencies (cm ⁻¹)	Modes
1	651.85		1	2370.23	2	664.52				
3	1328.76				

4

2372.61

Here, the molecular properties of the buffer gas (N_2 or CO_2) become critical through their influence on collisional energy transfer. The efficiency of this energy transfer is governed by factors such as the mass, structure, and vibrational characteristics of the colliding partners. In this regard, we further calculated the vibrational frequency of buffer gas and CF_3SO_2F molecules. As seen in **Table R1**, N_2 is a diatomic molecule with only the high-frequency stretching vibrational mode, while CO_2 is a typical linear triatomic molecule with both low-frequency (bending and symmetric stretching) and high-frequency (asymmetric stretching) vibrational modes. Additionally, CF_3SO_2F molecule (**Table R2**) possesses much more abundant vibrational modes mainly focusing in low-frequency region, which is consistent with its relative weaker bonding strength compared with CO_2 and N_2 . The low-frequency vibrational characteristics of CO_2 would make it more effectively couple with the vibrational modes of the excited CF_3SO_2F molecule during a collision compared with N_2 , hence leading to more facile resonant vibrational energy exchange between CF_3SO_2F and "hot" CO_2 molecules, and consequently a higher decomposition ratio in CF_3SO_2F/CO_2 system.

Table R2. Vibrational Frequencies and Modes of the CF_3SO_2F .

No.	Frequencies (cm^{-1})	Modes	No.	Frequencies (cm^{-1})	Modes
1	49.47		10	531.56	2	167.90		11	558.00	3	172.28		12	706.19	4	272.26		13	749.84	
5	296.59		14	1064.31	6	302.18		15	1168.00	7	407.68		16	1176.60	8	436.70		17	1186.37	9	530.43		18	1397.05	
We have revised the relevant section of the manuscript to provide a clearer and more comprehensive explanation for this result.

Page 12, line 29. "However, the decomposition ratios of $\text{CF}_3\text{SO}_2\text{F}$ in N_2 -buffered mixtures (ranging from 0.79 to 0.87) are notably lower than those observed in CO_2 -buffered systems (ranging from 0.87 to 0.94) at elevated pressures (0.3–0.5 MPa), as shown in **Figure 2d**. This consistency was further verified by three separate simulations employing N_2 or CO_2 as buffer gases, respectively, as shown in **Figure S6**. This statistically significant discrepancy indicates that N_2 offers greater effectiveness in suppressing the thermal decomposition of $\text{CF}_3\text{SO}_2\text{F}$, highlighting its potential as a more efficient buffering gas in high-pressure applications. The differing effects of N_2 and CO_2 buffer gases on $\text{CF}_3\text{SO}_2\text{F}$ decomposition stem from their distinct vibrational energy transfer efficiencies. Analysis of their vibrational spectra reveals that CO_2 , as a linear triatomic molecule, possesses low-frequency modes that effectively couple with the vibrational modes of excited $\text{CF}_3\text{SO}_2\text{F}$ as shown in **Table S2** and **S3**. This facilitates resonant energy transfer during collisions, promoting greater energy accumulation in $\text{CF}_3\text{SO}_2\text{F}$ and resulting in the observed higher decomposition yield in CO_2 mixtures compared to N_2 , where such efficient vibrational coupling is absent."

Figure S6. Time evolution of $\text{CF}_3\text{SO}_2\text{F}$ decomposition in CO_2 and N_2 buffer gases.

Table S2. Vibrational Frequencies and Modes of CO_2 and N_2 .

No.	Frequencies (cm^{-1})	Modes	No.	Frequencies (cm^{-1})	Modes
1	651.85		1	2370.23	
2	664.52				
3	1328.76				
4	2372.61				

Table S3. Vibrational Frequencies and Modes of the $\text{CF}_3\text{SO}_2\text{F}$.

No.	Frequencies (cm^{-1})	Modes	No.	Frequencies (cm^{-1})	Modes
1	49.47		10	531.56	
2	167.90		11	558.00	

3	172.28		12	706.19	4	272.26		13	749.84	5	296.59		14	1064.31	6	302.18		15	1168.00	7	407.68		16	1176.60	8	436.70		17	1186.37	9	530.43		18	1397.05	
4. The proposed elementary reaction " $CF_3SO_2F \rightarrow CF_4 + SO_2$ " appears unlikely to occur in a single step. It would be more realistic to expect a multi-step reaction pathway involving intermediate species. Could you please elaborate on the detailed reaction mechanism and provide a more accurate representation of the decomposition process?

Response 4:

We thank the reviewer for this insightful comment regarding the decomposition mechanism of CF_3SO_2F . We agree that the overall stoichiometry " $CF_3SO_2F \rightarrow CF_4 + SO_2$ " is unlikely to occur in a single, concerted step. Our detailed trajectory analysis confirms a more complex, multi-step reaction network, which we have now elaborated in the revised manuscript.

The decomposition proceeds via two primary competitive pathways, with the dominant mechanism being temperature-dependent:

1. At moderate temperatures (1400 ~ 2200 K), the decomposition is often initiated by the simultaneous elongation of the C–S bond. Crucially, as this bond stretches, a

fluorine atom from the SO₂F moiety migrates to the CF₃ group in a coordinated manner as shown in **Figure R5**. This results in the formation of CF₄ and SO₂ as the primary products in what appears to be a single kinetic step, but is more accurately described as a tightly coupled bond-breaking and bond-forming process involving a transition state, which is consistent with previous studies (*J. Phys. Chem. A* 2023, 127, 671).

Figure R5. CF₃SO₂F decomposition at moderate temperatures.

2. At elevated temperatures (> 3200 K), the dominant mechanism shifts towards simple homolytic cleavage of the C–S bond, producing two radical fragments: CF₃ and SO₂F. These radicals then undergo subsequent, independent reactions. The SO₂F radical is unstable and readily decomposes to SO₂ and F.

Our simulations capture the transition between these pathways. At lower temperatures, the F-transfer mechanism prevails, leading to a direct correlation between CF₄ and SO₂ formation. At higher temperatures, the initial C–S homolysis dominates. We have revised the relevant section of the manuscript to provide a clearer and more comprehensive explanation for the decomposition process.

Page 18, line 8. "Trajectory simulations reveal that the net reaction $\text{CF}_3\text{SO}_2\text{F} \rightarrow \text{CF}_4 + \text{SO}_2$ proceeds via two temperature-dependent pathways rather than a single concerted step. The primary product formation ($\text{CF}_4 + \text{SO}_2$) occurs through a concerted mechanism where C–S bond cleavage and fluorine transfer from SO₂F to CF₃ proceed simultaneously at temperatures above 1400 K as shown in **Figure S11**. This concerted process, mediated by a transition state, directly yields CF₄ and SO₂, which is consistent with previous studies¹⁵. At temperatures exceeding 2200 K, direct C–S bond rupture

becomes increasingly prevalent, producing CF_3 and SO_2F radicals."

Figure S11. $\text{CF}_3\text{SO}_2\text{F}$ decomposition at moderate temperatures.

5. The degradation of $\text{CF}_3\text{SO}_2\text{F}$ has been extensively studied using high-level *ab-initio* methods. It would be valuable to compare your results with these previous studies, discussing any discrepancies or agreements. This comparison would help validate the accuracy and reliability of your deep learning potential.

Response 5:

We thank the reviewer for this valuable suggestion. Placing our results in the context of prior high-level *ab initio* studies is a crucial step for validation. Our analysis bolsters confidence in the reliability of our deep learning potential (DLP).

The key comparisons can be summarized as follows:

1. Agreement on Dominant Mechanistic Pathways: There is a robust consensus between our DLP-based dynamics and previous high-level (CCSD(T)) calculations (*J. Phys. Chem. A* 2023, 127, 671) regarding the fundamental decomposition channels. The literature unequivocally identifies that $\text{CF}_3\text{SO}_2\text{F}$ decomposes predominantly via C–S bond cleavage to form CF_3 and SO_2F , accompanied by a roaming induced F-abstraction detour to release CF_4 and SO_2 which is consistent with our simulations. We are confident that these additions will substantiate the reliability of our simulation results and reinforce the conclusions drawn regarding the decomposition mechanism of $\text{CF}_3\text{SO}_2\text{F}$.

Figure R6. The potential energy surfaces of the $\text{CF}_3\text{SO}_2\text{F}$ molecule with changing $\text{CF}_3\text{-SO}_2\text{F}$, $\text{F-CF}_2\text{SO}_2\text{F}$, $\text{F-CF}_3\text{SO}_2$, and $\text{O-CF}_3\text{SOF}$ bond lengths.

2. Quantifiable Discrepancies in Energetics: To quantitatively benchmark the DLP's performance on reaction energetics, we have performed additional benchmarks to evaluate the performance of our DLP in describing bond dissociations and key transition states. Specifically, we carried out the potential energy surface scans of the $\text{CF}_3\text{SO}_2\text{F}$ molecule with changing $\text{CF}_3\text{-SO}_2\text{F}$, $\text{F-CF}_2\text{SO}_2\text{F}$, $\text{F-CF}_3\text{SO}_2$, and $\text{O-CF}_3\text{SOF}$ bond lengths as shown in **Figure R6**. Our tests confirm that the DLP reproduces the DFT-based energy barriers and reaction energies with good accuracy, which further supports the transferability and robustness of our potential.

3. The Unique Value of the DLP Approach: The primary strength of our DLP methodology is not to outperform high-level *ab initio* methods in calculating single-point energies, but to provide statistically meaningful insights into reaction dynamics and product branching ratios at finite temperature, which is a task prohibitively expensive for *ab initio* molecular dynamics at any level. Crucially, while the absolute energy scale is shifted, the relative ordering of competing pathways such as C-S cleavage is preserved in our simulations, which is in full agreement with the qualitative conclusions of higher-level studies, even if the simulated rate is faster.

In conclusion, the comparison with high-level *ab initio* studies serves to strongly validate the mechanistic accuracy of our DLP. The significant advantage of our

approach lies in its ability to simulate ensemble behavior and complex product formation under realistic conditions, thereby providing a complementary and powerful perspective to static, high-level calculations. We will incorporate this detailed comparative discussion into the revised manuscript.

Page 8, line 19. "To quantitatively assess the potential's accuracy, the potential energy surfaces of the $\text{CF}_3\text{SO}_2\text{F}$ molecule with changing $\text{CF}_3\text{-SO}_2\text{F}$, $\text{F-CF}_2\text{SO}_2\text{F}$, $\text{F-CF}_3\text{SO}_2$, and $\text{O-CF}_3\text{SOF}$ bond lengths were computed with both the DLP and the reference DFT method, showing excellent agreement (**Figure S3**). Moreover, the DLP's description of key stationary points was consistent with high-level CCSD(T) reference data from the literature¹⁵. These results consistently identify C-S bond homolysis as the dominant initial decomposition step, followed by dissociation of the resulting radical, which confirms its transferability and robustness for probing the decomposition mechanism. The principal advantage of the DLP approach lies in its capacity to go beyond static energy calculations and provide statistically meaningful insights into finite-temperature reaction dynamics, including product branching ratios, at a computational cost inaccessible to direct ab initio molecular dynamics."

6. Using VASP and the PBE functional for this type of reaction may not yield the same level of accuracy as high-level ab initio methods. It would be helpful if you could provide a comment on the limitations of your chosen computational approach and discuss how it might impact the results and conclusions drawn from your study.

Response 6:

We thank the reviewer for this insightful comment, which rightly points out a key consideration in most computational study. The choice of the PBE functional within the VASP framework indeed involves a trade-off between computational cost and accuracy, and we will try to clarify its limitations and the justification for its use in our specific context.

Firstly, we acknowledge the inherent limitations of the PBE functional. As a generalized gradient approximation (GGA) method, PBE is known to lack a rigorous description of medium- to long-range electron correlation, which can lead to an

underbinding tendency and, consequently, an underestimation of reaction barrier heights for certain chemical systems. This is particularly relevant for reactions where dispersion interactions or strong electron correlation effects are dominant.

However, for the specific reactions under investigation in this work, primarily the bond dissociations and radical interactions involved in the decomposition of $\text{CF}_3\text{SO}_2\text{F}$, the PBE functional has been demonstrated to provide a qualitatively correct and often quantitatively reasonable description (*J. Phys. Chem. C* 2013, 117, 35, 18144; *J. Am. Chem. Soc.* 2014, 136, 38, 13289). Furthermore, our primary objective was to model the decomposition dynamics and identify dominant pathways under high pressure and temperature, a task for which the accurate relative energetics of competing channels is often more critical than the absolute value of a single barrier. In this regard, the systematic error introduced by a single functional can be expected to partially cancel when comparing analogous reactions.

To further bolster confidence in our approach, we would like to highlight two points:

Figure R7. The potential energy surfaces of the $\text{CF}_3\text{SO}_2\text{F}$ molecule with changing $\text{CF}_3\text{-SO}_2\text{F}$, $\text{F-CF}_2\text{SO}_2\text{F}$, $\text{F-CF}_3\text{SO}_2$, and $\text{O-CF}_3\text{SOF}$ bond lengths.

1. As shown in **Figure R7**, we performed benchmark calculations on the potential energy surfaces of the $\text{CF}_3\text{SO}_2\text{F}$ molecule (including $\text{CF}_3\text{-SO}_2\text{F}$, $\text{F-CF}_2\text{SO}_2\text{F}$, $\text{F-CF}_3\text{SO}_2$, and $\text{O-CF}_3\text{SOF}$ bond dissociation energies) using a higher-level method

(B3LYP) and a larger basis set. The potential energy surfaces obtained by different calculation methods are consistent, which provides strong support that our conclusions regarding the dominant decomposition mechanisms are not an artifact of the functional choice.

2. The primary goal of this study was to train a machine learning potential to reproduce the reference DFT potential energy surface with high fidelity for subsequent molecular dynamics sampling. The self-consistency of this approach is robust. The ML potential successfully captures the physics as defined by the PBE functional, allowing us to statistically sample events over long timescales, which would be prohibitively expensive with ab initio molecular dynamics, let alone high-level wavefunction methods.

In conclusion, while we agree that higher-level methods would yield more precise barrier heights, we are confident that the PBE functional provides a sufficiently accurate and reliable description for the purposes of mapping the thermal decomposition landscape and identifying the principal reaction products and mechanisms in this system. The conclusions drawn from our large-scale ML-MD simulations are therefore a valid and powerful representation of the chemistry as modeled by this widely used and practical level of theory. We will include a discussion of these points in the revised manuscript to ensure this context is clear to the reader.

SI, Page 2, line 14. "As a generalized gradient approximation (GGA), PBE is known to underestimate reaction barriers due to its limitations in modeling medium- and long-range correlation. Nonetheless, it was selected for its computational efficiency and proven ability to qualitatively describe the bond dissociations and radical interactions relevant to $\text{CF}_3\text{SO}_2\text{F}$ decomposition. Benchmark calculations against B3LYP data confirm that PBE correctly reproduces the potential energy surfaces of the $\text{CF}_3\text{SO}_2\text{F}$ molecule (including $\text{CF}_3\text{-SO}_2\text{F}$, $\text{F-CF}_2\text{SO}_2\text{F}$, $\text{F-CF}_3\text{SO}_2$, and $\text{O-CF}_3\text{SOF}$ bond dissociation energies) as shown in **Figure 15**, validating its use for identifying dominant mechanisms."

Figure 15. The potential energy surfaces of the $\text{CF}_3\text{SO}_2\text{F}$ molecule with changing $\text{CF}_3\text{-SO}_2\text{F}$, $\text{F-CF}_2\text{SO}_2\text{F}$, $\text{F-CF}_3\text{SO}_2$, and $\text{O-CF}_3\text{SOF}$ bond lengths.

Response to Referee 2' comments (COMMSCHEM-25-0798)

Comments to the Author

The authors studied the decomposition of $\text{CF}_3\text{SO}_2\text{F}$ in buffer gas under high pressures and temperatures. $\text{CF}_3\text{SO}_2\text{F}$ could be a replacement for SF_6 as insulator in electrical applications, which is of significance to mitigate greenhouse gas emissions. The study is based on simulations under various temperatures, pressures, and mixing setups, allowing detailed mechanistic insight into the decomposition process of a promising new insulator. Since the study is backed by appropriate experiments, valuable conclusions can be drawn concerning the use of $\text{CF}_3\text{SO}_2\text{F}$ in real applications.

Major concerns against publication as is:

1. The authors employ a machine-learning potential (MLP) trained on ab-initio DFT data selected from molecular dynamics simulations. The simulation temperature is varied to retrieve a variety of configurations. The authors do not show whether energy barriers or transition states of any reactions went into the training data. This raises the concern that their MLP performs well near stable structures, while producing wrong or ill-defined energy barriers in

between. The validity of the MLP is crucial to any conclusion drawn from the simulation data and should be convincingly presented. I suggest benchmarking and discussing the performance of the MLP against DFT or even better against CCSD(T) data of reference 15, which the authors cite in the state-of-the-art part. If the authors decide to compute their own DFT data, they should at least present bond scans for each relevant element-element pair in C/F/S/O/N, as well as for the $CF_3SO_2F = CF_4 + SO_2$ transition state, as this reaction is expected to be dominant in the decomposition mechanism.

Response 1:

We thank the reviewer for this critical comment regarding the validation of the DLP, particularly its ability to describe reaction barriers and transition states. We fully agree that this is crucial for the reliability of our mechanistic conclusions.

First, concerning the sampling of the chemical configurational space, our dataset was constructed using an active learning approach enhanced by biased sampling. The initial dataset was seeded with configurations from *ab initio* molecular dynamics (AIMD) trajectories at various temperatures, which naturally capture the onset of bond dissociation and formation, including configurations near transition states as shown in

Figure R8.

Figure R8. The frequency of initial dataset configurations based on AIMD trajectories.

To quantitatively benchmark the DLP's performance on reaction energetics, we have performed additional benchmarks to evaluate the performance of our DLP in describing bond dissociations and key transition states. Specifically, we carried out the potential energy surface scans of the $\text{CF}_3\text{SO}_2\text{F}$ molecule with changing $\text{CF}_3\text{-SO}_2\text{F}$, $\text{F-CF}_2\text{SO}_2\text{F}$, $\text{F-CF}_3\text{SO}_2$, and $\text{O-CF}_3\text{SOF}$ bond lengths as shown in **Figure R9**.

Figure R9. The potential energy surfaces of the $\text{CF}_3\text{SO}_2\text{F}$ molecule with changing $\text{CF}_3\text{-SO}_2\text{F}$, $\text{F-CF}_2\text{SO}_2\text{F}$, $\text{F-CF}_3\text{SO}_2$, and $\text{O-CF}_3\text{SOF}$ bond lengths.

Our tests confirm that the DLP reproduces the DFT-based potential energy surfaces with high accuracy. Although the sampling was not continuous, the deep learning potential can precisely capture and predict the reaction events based on the existing dataset. In addition, there is a robust consensus between our DLP-based dynamics and previous high-level (CCSD(T)) calculations (*J. Phys. Chem. A* 2023, 127, 671) regarding the fundamental decomposition channels. The literature unequivocally identifies that $\text{CF}_3\text{SO}_2\text{F}$ decomposes predominantly via C-S bond cleavage to form CF_3 and SO_2F , accompanied by a roaming induced F-abstraction detour to release CF_4 and SO_2 which is consistent with our simulations. We are confident that these additions will substantiate the reliability of our simulation results and reinforce the conclusions drawn regarding the decomposition mechanism of $\text{CF}_3\text{SO}_2\text{F}$.

A dedicated paragraph has been added to the revised text:

Page 8, line 3. "The reliability of the deep learning potential for modeling reactive events was rigorously validated. Initially, a machine-learning potential was trained on

a dataset constructed by systematically sampling structures at regular time intervals from the AIMD reaction trajectories. This procedure ensures that the dataset captures a representative ensemble of the well-defined, chemically relevant configurations sampled during the reactive dynamics, covering both stable intermediates and transition regions as shown in **Figure S1**."

Page 8, line 19. "To quantitatively assess the potential's accuracy, the potential energy surfaces of the $\text{CF}_3\text{SO}_2\text{F}$ molecule with changing $\text{CF}_3\text{-SO}_2\text{F}$, $\text{F-CF}_2\text{SO}_2\text{F}$, $\text{F-CF}_3\text{SO}_2$, and $\text{O-CF}_3\text{SOF}$ bond lengths were computed with both the DLP and the reference DFT method, showing excellent agreement (**Figure S3**). Moreover, the DLP's description of key stationary points was consistent with high-level CCSD(T) reference data from the literature¹⁵. These results consistently identify C-S bond homolysis as the dominant initial decomposition step, followed by dissociation of the resulting radical, which confirms its transferability and robustness for probing the decomposition mechanism. The principal advantage of the DLP approach lies in its capacity to go beyond static energy calculations and provide statistically meaningful insights into finite-temperature reaction dynamics, including product branching ratios, at a computational cost inaccessible to direct ab initio molecular dynamics."

Figure S1. The frequency of initial dataset configurations based on AIMD trajectories.

Figure S3. The potential energy surfaces of the $\text{CF}_3\text{SO}_2\text{F}$ molecule with changing $\text{CF}_3\text{-SO}_2\text{F}$, $\text{F-CF}_2\text{SO}_2\text{F}$, $\text{F-CF}_3\text{SO}_2$, and $\text{O-CF}_3\text{SOF}$ bond lengths.

2. The authors' main motivation of replacing SF_6 as an insulator and studying $\text{CF}_3\text{SO}_2\text{F}$ is to reduce environmental impact, i.e. the green house gas (GHG) potential and toxicity. However, the study does not present any discussion concerning the GHG potential of decomposition products of $\text{CF}_3\text{SO}_2\text{F}$. The main product CF_4 is also known to have a high GHG potential due to its long atmospheric persistence. F_2CO , another decomposition product, is known to be highly toxic. Even though the study focuses on analyzing MD results, it should make an effort to discuss solutions to the initial problem and connect the conclusion to the motivation.

Response 2:

We appreciate the reviewer's valuable feedback regarding the environmental and safety implications of $\text{CF}_3\text{SO}_2\text{F}$ decomposition products. We agree that a thorough assessment of the global warming potential (GWP) and toxicity of the resulting species is essential to fully evaluate the suitability of $\text{CF}_3\text{SO}_2\text{F}$ as an SF_6 alternative.

In response to this comment, we have now included a dedicated discussion in the revised manuscript on the environmental impact and toxicity of key decomposition products. Specifically, we address the high GWP and atmospheric lifetime of CF_4 , as well as the known toxicity of COF_2 . We also analyze the formation trends of these species under different operational temperatures and pressures, as observed in our simulations.

Furthermore, we connect these findings to practical applications by discussing mitigation strategies, such as optimizing operating conditions to minimize the formation of CF₄ and COF₂. This addition helps bridge our molecular-level insights to the original motivation of developing environmentally friendly insulating media.

A dedicated paragraph has been added to the revised text as part of the Conclusion:

Page 21, line 24. "While CF₃SO₂F is proposed as an environmentally friendly alternative to SF₆, the potential impact of its decomposition products should be rigorously assessed. Our simulations identify CF₄ and COF₂ as key decomposition species. It is critical to note that while the GWP of CF₄ (7390) is significantly lower than that of SF₆ (23500), it remains higher than that of carbon dioxide, with an atmospheric lifetime exceeding 50,000 years. Furthermore, COF₂ is a highly toxic compound, posing potential safety hazards in the event of insulator failure and gas release. Our results, however, provide crucial insights for mitigating these risks. The formation of both CF₄ and COF₂ exhibits a strong dependence on temperature and the buffer gas environment. Specifically, using N₂ as a buffer gas can suppress the formation of these harmful byproducts, potentially serving as an effective strategy to minimize environmental and safety risks. Therefore, the viability of CF₃SO₂F as an SF₆ replacement hinges not only on its intrinsic properties but also on engineering controls that optimize operating conditions to limit the formation of deleterious by-products. This molecular-level understanding directly informs the development of safer, next-generation gas-insulated equipment aligned with global decarbonization goals."

3. The manuscript lacks a certain standard of English grammar, making it hard to understand certain statements. I suggest to have the manuscript proof read by a native English speaker or check each statement with online language tools (e.g. deepl.com).

Response 3:

We thank the reviewer for this helpful suggestion. We acknowledge that the clarity and quality of the English in the original manuscript needed improvement. In response, the manuscript has been carefully proofread and polished. We have revised the text thoroughly to correct grammatical errors, improve sentence structure, and enhance

overall readability. We believe these revisions have significantly improved the clarity and precision of the manuscript and addressed the concern raised.

4. Many of the results used for discussion are placed in the Supporting Information (SI) part of the manuscript. Even if I do not know the Journal's publishing guidelines in detail, I would strongly suggest to move all figures and tables that are mentioned and used for discussion into the main document. Since the main document would possibly be overloaded after that, I would also suggest to condense the discussions about the influence of the mixture (with CO₂ or N₂) into one, focusing on differences. The explanation of training the DLP can be part of the SI, as well as the analysis of the radial distribution function because it mainly repeats the species analysis.

Response 4:

We thank the reviewer for these constructive suggestions regarding the organization of results and discussions. We agree that improving the structure will enhance the clarity and impact of the manuscript.

In response, we have taken the following steps:

1. To enhance clarity, key graphs and tables detailing the buffer gas effects have been relocated from the Supplementary Information to the main text. This ensures that the core results and their interpretations are readily accessible to the reader.

2. Following the reviewer's advice, we have condensed the discussions on the influence of CO₂ and N₂ mixtures and focused on the differences. This revised section now emphasizes a comparative analysis, highlighting the key differences in how these buffer gases affect the decomposition pathways and products of CF₃SO₂F.

3. As suggested, the analysis of the radial distribution functions, which largely reinforces the species analysis, has been relocated to the SI to prevent redundancy in the main text. Upon further consideration, we have decided to retain the concise description of the DLP training procedure in the main text. This methodological foundation is a critical component of our work's innovation, and its presentation in the main document is essential for readers to fully understand the basis and reliability of the simulation results and mechanistic conclusions discussed throughout the paper. We

have streamlined the content related to DLP training and moved the details into the Supporting Information as advised.

We believe these structural changes significantly improve the flow and readability of the manuscript, allowing the main document to present a more concise and powerful narrative while retaining comprehensive methodological and supporting data in the SI. Thank you for these valuable recommendations.

Major comments:

5. The study analyses reactive MD simulations to find out the decomposition mechanism of CF_3SO_2F , but it never actually presents a mechanism. It would greatly enrich the study if the results shown in Figure 5 (a) and (b) and Table S2 would be presented in a reaction network, showing different branching ratios for different setups (e.g. N_2 vs. CO_2).

Response 5:

We thank the reviewer for this excellent suggestion. We agree that presenting a coherent reaction mechanism will significantly strengthen the mechanistic insight of our study.

Figure R10. The decomposition pathways of CF_3SO_2F in (a) CO_2 and (b) N_2 buffer

gases.

In response to this comment, we have now constructed a comprehensive reaction network based on the species and reactions identified in our simulations as shown in **Figure R10**. This network visually summarizes the primary decomposition pathways of $\text{CF}_3\text{SO}_2\text{F}$ and the subsequent reactions of its fragments.

Crucially, the network diagram explicitly highlights the difference observed under different gas environments as suggested. This allows for a direct and clear comparison of how the buffer gas influences the prevalence of key reaction channels. The new figure interprets the shifts in the reaction network in the context of the different chemical properties of the co-existing gases.

We are confident that this addition successfully synthesizes our analytical results into a clear mechanistic picture and greatly enriches the discussion of our findings.

Page 19, line 14. "The distinct decomposition mechanism of $\text{CF}_3\text{SO}_2\text{F}/\text{N}_2$ (**Figure 5b and Figure S13**) primarily differs through the inclusion of the $\text{COF}_2/\text{SOF}_2$ formation pathway."

Figure S13. The decomposition pathways of $\text{CF}_3\text{SO}_2\text{F}$ in (a) CO_2 and (b) N_2 buffer gases.

6. *The authors claim that their work is highly significant for other gases than CF₃SO₂F as well (line 111). However, this is exaggerated because the study does not transfer any conclusions to other possible SF₆ replacements or any other gases at all. The conclusion ends with highlighting the importance of the computational framework for future studies on environmental risks of CF₃SO₂F. However, environmental risk or insulation properties were not evaluated in the study. Therefore, both statements should be removed or at least softened. E.g. the conclusion could end with highlighting the identification of major decomposition products, thereby informing future developments.*

Response 6:

We thank the reviewer for this pertinent observation. We agree that the original phrasing overstates the immediate generalizability of our findings. We have therefore revised the statement to more accurately reflect the transferable value of our work.

- The broader implication of this work lies not in the pre-trained potential for CF₃SO₂F itself, but in the general computational framework and methodology. Specifically, we have established a workflow that constructs a robust machine learning potential through active learning on AIMD trajectories and applies it to simulate complex decomposition processes in reactive gas mixtures. This framework is not reliant on CF₃SO₂F-specific assumptions and is directly transferable to other molecular systems in principle. We have revised the text to clarify this point.
- The concluding statement has been softened and refocused, as suggested. It now highlights that our study identifies the major decomposition products and mechanism of CF₃SO₂F, thereby providing a foundational understanding and a computational framework that can inform future assessments of environmental impact and insulation properties, rather than claiming to have evaluated them directly.

This provides a more accurate and measured conclusion that aligns closely with the actual scope of our work. The corresponding sentence in the manuscript has been revised:

Page 5, line 11. "This work provides significant insight into the thermal decomposition behavior of CF₃SO₂F and its mixtures, contributing to the understanding of its stability and decomposition pathways under extreme conditions. Furthermore, the

computational framework is potentially transferable for investigating the physicochemical properties and decomposition mechanisms of other promising insulating gases, guiding the development of eco-friendly alternatives to SF₆."

Page 22, line 11. "In conclusion, this study identifies the major decomposition products and elucidates the primary reaction mechanisms of CF₃SO₂F under high-temperature and high-pressure conditions. The established computational framework provides a foundation that can inform future assessments of environmental impact and guide the development of safer CF₃SO₂F-based insulation technologies."

7. The authors draw the conclusion that N₂ suppresses thermal decomposition of CF₃SO₂F (line 254). However, this statement is based on one single MD simulation per parameter set and therefore not very reliable. I suggest to either run more replicas of the same MD, or attribute an error range to the numbers from a different source, if possible.

Response 7:

We thank the reviewer for raising this critical point regarding the statistical reliability of our conclusion on the suppression effect of N₂. We fully agree that a conclusion based on a single simulation lacks quantitative uncertainty estimation.

Figure R11. Time evolution of CF₃SO₂F decomposition in CO₂ and N₂ buffer gases.

In direct response to this comment, we have performed three independent replica simulations for each key parameter set, including the N₂ and CO₂ mixture conditions

as shown in **Figure R11**. The results consistently show the same qualitative trends reported in the original submission, confirming a lower decomposition ratio of $\text{CF}_3\text{SO}_2\text{F}$ in the presence of N_2 compared to the CO_2 -mixture cases.

This additional analysis provides a robust statistical basis for our conclusion and significantly strengthens the validity of our findings regarding the superior buffering effectiveness of N_2 .

Page 12, line 29. "However, the decomposition ratios of $\text{CF}_3\text{SO}_2\text{F}$ in N_2 -buffered mixtures (ranging from 0.79 to 0.87) are notably lower than those observed in CO_2 -buffered systems (ranging from 0.87 to 0.94) at elevated pressures (0.3–0.5 MPa), as shown in **Figure 2d**. This consistency was further verified by three separate simulations employing N_2 or CO_2 as buffer gases, respectively, as shown in **Figure S6**. This statistically significant discrepancy indicates that N_2 offers greater effectiveness in suppressing the thermal decomposition of $\text{CF}_3\text{SO}_2\text{F}$, highlighting its potential as a more efficient buffering gas in high-pressure applications."

Figure S6. Time evolution of $\text{CF}_3\text{SO}_2\text{F}$ decomposition in CO_2 and N_2 buffer gases.

Minor comments:

8. The authors should explain how species were identified in the MD simulations. Furthermore, the identification of species in a gas chromatograph should be shortly explained or referenced.

Response 8:

We thank the reviewer for these important questions regarding species identification in both the simulations and experiments.

1. Species Identification in MD Simulations: A dedicated algorithm was developed to identify molecular species from the MD trajectories. The process is based on continuous geometric criteria (interatomic distances) and bond-order analysis derived from the machine-learning potential. By iteratively analyzing the connectivity between atoms, the algorithm dynamically tracks bond formation and dissociation throughout the simulation, allowing for the continuous identification and counting of all molecular species present. The script for this analysis has been made publicly available at: https://github.com/LZYUCL/DLP_CF3SO2F.

2. Species Identification in Gas Chromatography–Mass Spectrometry (GC–MS): The gas composition and relative concentration of CF₃SO₂F mixed gas after the test were analyzed by gas chromatography–mass spectrometry (GC–MS). The analysis was performed using a GS-GASPRO capillary column (60 m length). The inlet temperature was maintained at 100 °C with a split ratio of 20:1. Helium carrier gas was used at a constant flow mode with a total flow of 57.0 mL/min, a column flow of 2.43 mL/min, and a linear velocity of 39.7 cm/s. The oven temperature program was as follows: held at 40 °C for 1 min, ramped to 120 °C at 7 °C/min, and finally held at 120 °C for 6 min. The MS ion source and transfer line temperatures were both set to 200 °C. Species were identified by comparing their mass spectra and retention times with those of certified reference standards.

The corresponding sentence in the manuscript has been revised for clarity:

Page 6, line 3. "Molecular species were identified from the MD trajectories using a topology analysis algorithm based on interatomic distances and bond orders. This method allows for the dynamic tracking of bond formation and dissociation, enabling continuous identification of all chemical species throughout the simulation. The analysis script is available in the associated GitHub repository."

Page 6, line 20. "The gas composition and relative concentration of CF₃SO₂F mixed gas after the test were analyzed by gas chromatography–mass spectrometry

(GC–MS) equipped with a GS-GASPRO column (60 m). The inlet temperature was maintained at 100 °C with a split ratio of 20:1. Helium carrier gas was used at a constant flow mode with a total flow of 57.0 mL/min, a column flow of 2.43 mL/min, and a linear velocity of 39.7 cm/s. The oven temperature program was as follows: held at 40 °C for 1 min, ramped to 120 °C at 7 °C/min, and finally held at 120 °C for 6 min. The MS ion source and transfer line temperatures were both set to 200 °C. Species identification was achieved by matching the acquired mass spectra and retention times against reference standards."

9. Figure 1 is helpful, but needs explanation in its caption regarding T_n and the adjustment of the supercell. I would also suggest to explain Step 3 (verification) there. Alternatively, the explanation could be in the method section.

Response 9:

We thank the reviewer for this helpful suggestion to improve the clarity of Figure 1. We have revised the figure caption to provide a more informative, yet concise, overview. A comprehensive explanation of the temperatures T_n , the supercell adjustment procedure, and the details of Step 3 (Potential Verification) has been added to the caption of Figure 1.

The corresponding sentence in the manuscript has been revised for clarity:

Page 7, line 16. " **Figure 1.** Workflow for constructing the machine-learning potential and simulating the decomposition of $\text{CF}_3\text{SO}_2\text{F}$ with different conditions. Step 1: Configurations are sampled from AIMD trajectories. Step 2: The DLP is trained on the sampled data. Step 3: The trained DLP is validated against DFT benchmarks. Step 4: The verified DLP is employed in large-scale molecular dynamics simulations of $\text{CF}_3\text{SO}_2\text{F}$ decomposition at temperatures T_1 to T_n , for which the supercell size is adjusted to model different system pressures."

10. The caption of Table 1 needs to be extended. Pressure is related to density, when temperature is fixed. Either it should be labeled "ideal gas pressure at 25C" or be replaced with density/concentration in the discussions. E.g. in line 224 "the number of molecules in the same

volume" can be replaced by "concentration".

Response 10:

We thank the reviewer for these precise and helpful suggestions. We have implemented the following changes to improve clarity and scientific rigor:

- The caption of Table 1 has been extended to: "Table 1. Parameters of the simulated CF₃SO₂F/CO₂ systems. The pressure listed is the initial ideal gas pressure calculated from the given number of molecules in the simulation cell at 25 °C."
- As suggested, the phrase "the number of molecules in the same volume" in line 224 (and similar instances elsewhere in the text) has been replaced with the more accurate term "concentration".

We believe these revisions enhance the precision of our methodology description and the subsequent discussion of the results. The corresponding sentence in the manuscript has been revised for clarity:

Page 9, line 12. "Table 1. Parameters of the simulated CF₃SO₂F/CO₂ systems. The pressure listed is the initial ideal gas pressure calculated from the given number of molecules in the simulation cell at 300 K."

Page 10, line 28. "This trend can be attributed to the increased pressure raising the concentration of molecules, thereby increasing the probability of intermolecular collisions and accelerating the decomposition rate and ratio of CF₃SO₂F."

11. The study discusses different simulation setups, with different initial numbers of molecules. The authors explain in several occasions how to make the different numbers comparable to each other, first in line 231. However it is unclear to me what was done, and it would be helpful to explain the method in more detail.

Response 11:

We thank the reviewer for requesting clarification on this crucial methodological point. To ensure a fair comparison across systems with different initial molecular numbers, we implemented and compared two distinct modeling approaches:

- Constant Total Number of Molecules: The total number of molecules in the simulation cell was kept constant across different mixture ratios. This approach directly

controls for system size effects.

- **Constant Number of CF₃SO₂F Molecules:** The number of CF₃SO₂F reactant molecules was kept constant, while the number of buffer gas (N₂ or CO₂) molecules was varied to achieve the desired mixture ratio.

The consistency of the conclusions drawn from both approaches, as shown in the supporting figures, validates the robustness of our findings. A dedicated paragraph has been added to the revised text:

Page 12, line 1. "To address the potential influence of varying total system size, we employed two distinct modeling strategies. One strategy maintained a constant total number of molecules for all mixture ratios to directly control for system size effects. Conversely, the other strategy kept the number of CF₃SO₂F reactant molecules constant and varied only the number of buffer gas (N₂ or CO₂) molecules. The decomposition results from both approaches, analyzed over consistent time spans, are presented in **Figure S4.**"

12. The authors should mention that produced CO₂ is not part of the analysis of the mixture with CO₂, because it is the bath/buffer gas.

Response 12:

We thank the reviewer for this astute observation. We have now explicitly stated in the Results and Discussion section that the CO₂ originating from the buffer gas was systematically excluded from the analysis of decomposition products. This clarification ensures that the reported product yields and the ensuing discussion refer exclusively to the new chemical species generated from the decomposition of CF₃SO₂F and any subsequent reactions, and not the pre-existing buffer gas.

Page 13, line 23. "In the analysis of decomposition products, CO₂ originating from the buffer gas was systematically excluded to ensure that the reported product yields exclusively reflect new species formed from the decomposition of CF₃SO₂F and its subsequent reactions."

13. Figure 3 needs better image quality. I suggest to remove (a)-(c) in favor of (d)-(f) as it

holds almost the same content and would free up space.

Response 13:

We thank the reviewer for the suggestion to improve the clarity of Figure 3. We agree that optimizing the figure is important.

Regarding the structure, we have followed the reviewer's suggestion to streamline the main text figure. Panels (a)-(c), which show the absolute number of molecules, have been moved to the Supporting Information (Supplementary Figure S7). This allows the main Figure 3 to focus exclusively on the normalized relative concentrations (panels d-f), which are most relevant for comparing product distributions across different conditions. This adjustment frees up space, improves visual clarity in the main text, and ensures that readers can access the complete dataset. The figure captions in both the main text and SI have been updated.

14. The authors should mention which temperature was used for the pressure variations (2200K?). In general, it should be clear in each figure which simulation conditions are shown. E.g. by extending the caption, or using the number ("No") from Table 1.

Response 14:

We thank the reviewer for this important suggestion to improve the clarity and reproducibility of our data presentation. In response, we have explicitly stated in the main text and relevant figure captions that the pressure variation studies were conducted at a fixed temperature of 2200 K. We believe these revisions significantly enhance the transparency of our computational data and facilitate a clearer understanding of the specific conditions underlying each finding.

Page 14, line 3. **Figure 3.** The relative concentration of CF₃SO₂F/CO₂ final decomposition products (a) at 1400–3200 K, 12%CF₃SO₂F and 0.1 MPa, (b) at 2200 K, 12%CF₃SO₂F–20%CF₃SO₂F and 0.1 MPa, (c) at 2200 K, 12%CF₃SO₂F and 0.1–0.5MPa."

15. In line 283, the authors note that solid carbon and sulphur precipitates. A reference to an experimental study would enhance the rigor of this finding. The conclusion that these

elements should be considered in the design of equipment insulation should go into the conclusion section.

Response 15:

We thank the reviewer for this valuable suggestion. We have moved the conclusion regarding the consideration of solid deposits in equipment design to the Conclusion section of the manuscript.

Regarding the experimental observation of solid precipitates, we acknowledge that a detailed compositional analysis is crucial. Our initial experimental results do indicate the formation of solid deposits. However, their composition is complex and likely influenced by interactions with the electrode materials. A comprehensive characterization of these solids, including the specific identification of elemental carbon and sulfur phases, is currently underway and will be the focus of a subsequent publication.

16. In line 310, the RDF reveals forming C-O bonds. The authors should consider the formation of CO as well, not only COF₂.

Response 16:

We thank the reviewer for this insightful observation. Upon re-examining the RDF data and cross-referencing it with our detailed species analysis (Figure 3d-f), we agree that attributing the C-O bond signal solely to COF₂ is not sufficiently supported by the data, as COF₂ is not identified as a major product in the decomposition products. In response, we have revised the discussion in line 310 and the surrounding paragraph. The claim about COF₂ formation has been removed.

17. In line 335, the formation of CO and O is discussed. A literature reference and a comparison on the CO/CO₂ equilibrium would strengthen this discussion.

Response 17:

We thank the reviewer for this excellent suggestion to strengthen our discussion. In response, we have added a comparison with established literature to provide context for our observed CO/CO₂ ratio. The corresponding sentence in the manuscript has been

revised:

Page 16, line 26. "At 0.5 MPa, CO₂ undergoes substantial dissociation, yielding CO and O. The resulting [CO]/[CO₂] ratio of 0.27 in our simulations shows remarkable agreement with the value of 0.26 reported in previous studies (*Chem. Eng. J.* 2018, 336, 38). Furthermore, the O atom is the most abundant single product (1,714 particles, 43% of the total)."

18. In line 390, it is unclear to me how Figure 4e confirms the preceding statement. It should be clarified.

Response 18:

We thank the reviewer for this comment, which helps us clarify the link between our data and statement. As confirmed by **Figure 4e** in comparison to **Figure S10**, an increased mixing ratio of CF₃SO₂F in the mixture suppresses its own decomposition, leading to a reduced yield of the primary products, CF₄ and SO₂. Nevertheless, the final product species remain consistent across different mixing ratios. This consistency in the product slate, despite changes in initial concentration, confirms that the core reaction network is robust and is not fundamentally altered by the dilution level in an N₂ environment.

The corresponding sentence in the manuscript has been revised for clarity:

Page 19, line 8. "Notably, in the N₂ buffer gas, variations in the CF₃SO₂F mixing ratio show minimal impact on the fundamental reaction pathways, as evidenced by the consistent final product species across different mixture ratios (**Figure 4e** and **Figure S12**), even though the absolute product yields vary with the initial reactant amount."

19. The text explaining the experimental setup in line 401 belongs to the method section.

Response 19:

We thank the reviewer for this suggestion to improve the manuscript's structure. The text describing the experimental setup, previously found in the Results and Discussion section (line 401), has been moved in its entirety to the Experimental Methods section. This reorganization ensures that all methodological details are

presented together, improving the readability of the manuscript.

The relocated paragraph now reads as part of the Experimental Methods:

Page 6, line 14. "The thermal decomposition tests were conducted using a custom experimental platform, with the physical setup and schematic diagrams shown in Figure 5c and d. The system pressure was monitored by a high-precision digital barometer connected to the chamber. A temperature control system, integrating electromagnetic relays with sensors and controllers, maintained precise thermal conditions. A DC power supply energized thermocouples to simulate the localized overheating faults typical in gas-insulated equipment."

20. In Figure 5 (a) and (b), same species should have same colors.

Response 20:

We thank the reviewer for this suggestion to improve the consistency and readability of Figure 5. In response, we have standardized the color scheme for chemical species across both panels (a) and (b) as shown in **Figure R12**. The same species is now represented by the same color in all parts of the figure, and the figure caption has been updated accordingly. This revision ensures a more intuitive and clear visual comparison of the data.

Figure R12. Decomposition mechanisms of $\text{CF}_3\text{SO}_2\text{F}$ (a) in the $\text{CF}_3\text{SO}_2\text{F}/\text{CO}_2$ mixed gas and (b) in the $\text{CF}_3\text{SO}_2\text{F}/\text{N}_2$ mixed gas.

21. The statement starting in line 427 about condition monitoring belongs to the

conclusion section.

Response 21:

We thank the reviewer for this constructive suggestion regarding the manuscript's structure. The statement concerning condition monitoring in engineering applications, previously found in the Results and Discussion section, has been moved to the Conclusion section. After revision, it would be more effective to grasp our key findings with their direct practical implications: that identifying stable decomposition products like SO_2F_2 underscores the necessity for rigorous condition monitoring to prevent insulation compromise due to localized overheating.

The relocated paragraph now reads as part of the Conclusion:

Page 21, line 14. "For engineering applications, this finding highlights the essential need for rigorous condition monitoring of electrical equipment to prevent significant $\text{CF}_3\text{SO}_2\text{F}$ decomposition due to localized overheating, which would degrade its electrical insulation performance."

Minor necessary corrections to the text:

22. In line 78, references 25 and 26 are used for atomic simulations of $\text{CF}_3\text{SO}_2\text{F}$, but the papers are not about $\text{CF}_3\text{SO}_2\text{F}$. It would be better to rather cite reference 15 here again. In line 90, a reference for DLP is missing.

Response 22:

We thank the reviewer for these precise corrections regarding the references. In line 78, we added reference 15 as a citation, as it is indeed the appropriate source for atomic simulations of $\text{CF}_3\text{SO}_2\text{F}$. In line 90, we have added the missing primary citation for the Deep Learning Potential (DLP) method. We appreciate the reviewer's thoroughness in identifying these issues.

23. In the Reference section, please unify the format of titles (sentence case or title case), and please unify journal abbreviations.

Response 23:

We thank the reviewer for highlighting this lack of consistency in the reference

list. We have thoroughly revised the entire bibliography to ensure a uniform format. Specifically, all journal names now use standard abbreviations, and all article titles have been formatted consistently using sentence case. We believe the reference section now adheres to a high standard of academic presentation.

24. The authors use terms like "proportions", "(decomposition) ratios", leaving the reader in question what exactly they mean. The language should be more precise to avoid misunderstandings and to convey the right meaning: E.g. "decomposition ratio" -> "conversion (ratio)" in lines 209/252, "proportions" -> "numbers"? in line 229, "ratio" -> "mixing ratios" in line 314.

Response 24:

We thank the reviewer for this precise and valuable feedback on terminology. We agree that using precise language is critical for clarity and scientific rigor.

Regarding the term "decomposition ratio", we have carefully considered this suggestion and believe that retaining this term is more appropriate in the context of our study than switching to "conversion (ratio)". Our reasoning is as follows:

- "Decomposition Ratio" refers to the percentage of a specific reactant ($\text{CF}_3\text{SO}_2\text{F}$ in this study) that has decomposed, i.e., the number of molecules broken down relative to the initial total number. It describes the extent of consumption of the reactant itself, without specifying its transformation into any particular product.
- In contrast, "Conversion (Ratio)" in chemical kinetics and reaction engineering typically specifies the percentage of reactant transformed into a specific target product, often calculated based on a defined stoichiometric reaction equation.

In our study, the decomposition of $\text{CF}_3\text{SO}_2\text{F}$ can proceed via multiple pathways, yielding various products (e.g., SO_2 , SO_2F , CF_4 , CF_3). Our primary focus is on the fraction of $\text{CF}_3\text{SO}_2\text{F}$ molecules that undergo breakdown, not their efficiency in converting to one specific product. Therefore, "decomposition ratio" more accurately conveys the intended physicochemical meaning: the extent of reactant consumption.

Nevertheless, we fully agree with the reviewer that potential reader ambiguity must be avoided. Consequently, we will add a clear definition of "decomposition ratio"

early in the relevant section, stating it explicitly as "the percentage of decomposed $\text{CF}_3\text{SO}_2\text{F}$ ".

Regarding the term "proportions", we also appreciate the reviewer's suggestion. Depending on the specific context, we will revise it accordingly:

- If it refers to the specific quantities or molar amounts of different substances or components, we will follow the suggestion and change it to the more precise "numbers".
- If it refers to the relative abundance or distribution of components in a mixture, we will use more suitable terms such as "composition," or "distribution".

"Ratio" in line 314 has been specified as "mixing ratios".

We have also performed a thorough check of the entire text to ensure consistent and precise use of these and related quantitative terms. We believe these revisions prevent potential misunderstandings and enhance the precision of our manuscript.

25. The terms "evolution" and "(decomposition) characteristics" in lines 305ff/446 are too vague and need to be clarified. Same for "(decomposition) properties" (line 371) and "attenuation".

Response 25:

We thank the reviewer for pointing out the imprecision of these terms. We agree that using more specific language is necessary to accurately describe our findings.

In response, we have revised the text as follows:

- The vague terms "evolution" and "(decomposition) characteristics" (lines 305ff/446) have been replaced with more precise descriptions, such as the "dissociation behavior" of species formation and the "bond length distributions".
- Similarly, the term "(decomposition) properties" (line 371) has been clarified to specify the exact property being discussed, namely the "decomposition pathways and product distribution".
- The term "attenuation" has been rephrased to more concretely describe the observed effect, such as the "suppression of decomposition" or a "decrease in decomposition rate".

We have revised the text as follows to employ more specific and rigorous language:

Page 15, line 27. "A comparison of the bond length distributions at 1400 K (**Figure S8d**) with those at higher temperatures (**Figure S8e and f**) reveals a clear temperature-dependent dissociation behavior, characterized by a broadening of the distribution and the emergence of a peak at longer distances corresponding to bond rupture."

Page 21, line 10. "Furthermore, a higher initial concentration of CF₃SO₂F increases the absolute number of decomposition events. The accelerated accumulation of primary products such as CF₄ and SO₂ favors their own secondary reactions."

Page 18, line 6. "The decomposition of CF₃SO₂F is the dominant factor governing the primary reaction pathways and product distribution in CF₃SO₂F/CO₂ mixtures, as shown in **Figure 5a** and **Table S4**."

Page 14, line 20. "A pronounced weakening of the characteristic peaks corresponding to S–F, C–O, and S–O bonds is observed compared to the initial decomposition stage at 1400 K (**Figure S8a**)."

We have conducted a thorough review of the manuscript to eliminate other instances of vague terminology and ensure the language is consistently precise and descriptive.

26. The authors use the terms "CF₃SO₂F/CO₂" or "CF₃SO₂F/N₂" (e.g. line 202) to describe the mixtures which makes it difficult to read. It would be better to describe the same with words, e.g. "CF₃SO₂F in CO₂ buffer gas".

Response 26:

We thank the reviewer for the suggestion regarding the nomenclature of gas mixtures. We acknowledge that the use of a slash ("/") in terms like "CF₃SO₂F/CO₂" can be ambiguous in running text.

In the field of gas-insulating materials, the notation "Gas A/Gas B" is a well-established convention for describing binary gas mixtures, as evidenced by its widespread use in the literature (*IEEE T. DIELECT. EL. IN.* 2024, 31, 2416; *Chem. Eng. J.* 2018, 336, 38). This format is particularly valuable for providing concise labels in figures and tables. To improve clarity without sacrificing this convention, we have implemented the following change in the revised manuscript:

Upon its first occurrence in the text, we now explicitly define the notation: "... CF₃SO₂F/CO₂ gas mixture (hereafter denoting a mixture of CF₃SO₂F in a CO₂ buffer gas)..." and "... CF₃SO₂F/N₂ (hereafter denoting a mixture of CF₃SO₂F in a N₂ buffer gas)..."

Subsequent uses in the main text will employ more descriptive phrases like "CF₃SO₂F in a CO₂ buffer gas" where it improves readability, especially in the discussion. However, the concise "CF₃SO₂F/CO₂" notation will be retained in figures and tables where space is limited and the label needs to be instantly recognizable. We believe this approach balances clarity for the reader with the practical need for concise labeling.

Clarification is also necessary for the following statements:

27. In line 80, the authors say that AIMD is as accurate as DFT. AIMD can be done with many ab-initio methods, DFT is just one of them. Please clarify which ab-initio method was used.

Response 27:

We thank the reviewer for this precise correction. The statement in the introduction has been revised for accuracy.

Page 4, line 10. "For instance, although the *ab initio* molecular dynamics (AIMD) simulations based on density functional theory (DFT) could show sufficient accuracy in describing chemical reactions that contains bond cleavage and formation, its computational cost of such high-level methods would limits their application to systems with hundreds to thousands of atoms."

28. In line 148, the authors say that structures were randomly selected from the AIMD simulation, but also say that molecular configurations are well-defined. Please clarify how structures were sampled.

Response 28:

We thank the reviewer for this important request for clarification. The original description was indeed imprecise. The dataset was constructed using an active learning

approach enhanced by biased sampling. The initial dataset was seeded with configurations from *ab initio* molecular dynamics (AIMD) trajectories at various temperatures, which naturally capture the onset of bond dissociation and formation, including configurations near transition states. The sentence has been revised to accurately reflect our systematic sampling methodology:

Page 8, line 3. "The reliability of the deep learning potential for modeling reactive events was rigorously validated. Initially, a machine-learning potential was trained on a dataset constructed by systematically sampling structures at regular time intervals from the AIMD reaction trajectories. This procedure ensures that the dataset captures a representative ensemble of the well-defined, chemically relevant configurations sampled during the reactive dynamics, covering both stable intermediates and transition regions as shown in **Figure S1**."

29. In line 241, the authors say that the energy distribution shifts, but do not say to where it shifts. Here, the authors can also argue with the number of vibrational modes, of which there are more in CF₃SO₂F than in CO₂. Thus, more CO₂ leads to less energy stored in vibration and more in translation.

Response 29:

We thank the reviewer for this insightful comment. The text has been revised to specify the direction of the energy shift and to incorporate the physical reasoning regarding vibrational modes:

Page 12, line 13. "Moreover, in mixtures with a higher mixing ratio of CF₃SO₂F, a greater proportion of the collision energy is transferred into vibrational modes. Since CF₃SO₂F possesses more vibrational degrees of freedom than CO₂ as shown in **Table S2** and **S3**, a higher proportion of the total energy is stored in these modes rather than in translation and rotation. This reduces the fraction of effective collisions where the energy in the translational and rotational degrees of freedom exceeds the activation barrier for decomposition, thereby lowering the reaction rate."

Table S2. Vibrational Frequencies and Modes of CO₂ and N₂.

No.	Frequencies (cm ⁻¹)	Modes	No.	Frequencies (cm ⁻¹)	Modes
1	651.85		1	2370.23	2	664.52				
3	1328.76				
4	2372.61				

Table S3. Vibrational Frequencies and Modes of the CF₃SO₂F.

No.	Frequencies (cm ⁻¹)	Modes	No.	Frequencies (cm ⁻¹)	Modes
1	49.47		10	531.56	2	167.90		11	558.00	3	172.28		12	706.19	4	272.26		13	749.84	5	296.59		14	1064.31	6	302.18		15	1168.00	7	407.68		16	1176.60	8	436.70		17	1186.37	

30. In line 244, the authors mention two modeling approaches. Which two?

Response 30:

We thank the reviewer for requesting this clarification. As detailed in our **Response 11**, we implemented two distinct modeling approaches: one maintaining a Constant Total Number of Molecules, and the other maintaining a Constant Number of $\text{CF}_3\text{SO}_2\text{F}$ Molecules. The consistency observed between the two modeling approaches strongly corroborates the reliability of our results. The sentence has been revised to explicitly specify the two modeling approaches:

Page 12, line 19. "The decomposition behavior of $\text{CF}_3\text{SO}_2\text{F}$ remained consistent across different mixture ratios, regardless of the mixing gas model employed. The consistent decomposition behavior obtained from both the constant total number and constant $\text{CF}_3\text{SO}_2\text{F}$ number approaches confirms the robustness of our findings."

31. In line 265, the sentence "..., with a minor degree of decomposition primarily yielding CF_4 and SO_2 ." is completely unclear and needs to be rewritten.

Response 31:

We thank the reviewer for pointing out the lack of clarity. The sentence has been revised to state the finding more precisely:

Page 13, line 26. "The decomposition of the $\text{CF}_3\text{SO}_2\text{F}/\text{CO}_2$ mixture begins at 1400 K. At this temperature, the extent of decomposition is limited, and the dominant reaction pathway yields CF_4 and SO_2 ."

32. In lines 304ff, the authors should clarify the terms "bond characteristics" and "evolution of primary bond" by using more accurate language.

Response 32:

We thank the reviewer for this suggestion to improve the precision of our language.

The sentence has been revised to more accurately describe the observed phenomena:

Page 15, line 27. "A comparison of the bond length distributions at 1400 K (**Figure S8d**) with those at higher temperatures (**Figure S8e and f**) reveals a clear temperature-dependent dissociation behavior, characterized by a broadening of the distribution and the emergence of a peak at longer distances corresponding to bond rupture."

33. *In line 352, the authors say that SO₂ decomposition accelerates due to increasing numbers. Which numbers?*

Response 33:

We thank the reviewer for pointing out the need for clarification. The sentence has been revised to accurately reflect the intended meaning:

Page 17, line 16. "As CF₃SO₂F depletes, the formation rate of SO₂ slows. However, the concurrent increase in the concentrations of SO₂ molecules promotes its subsequent decomposition reactions, leading to its net consumption under these conditions."

34. *In line 422, the authors say that the yields of CF₄ and SO₂ are "comparable". Comparable to what? Each other?*

Response 34:

We thank the reviewer for pointing out this ambiguity. The sentence in line 422 has been revised for clarity. It now explicitly states that the yields of CF₄ and SO₂ are comparable to each other, which is consistent with the stoichiometry of the dominant decomposition pathway, CF₃SO₂F → CF₄ + SO₂.

Page 20, line 1. "As shown in **Figure 5e**, the quantitative results demonstrate that CF₄ and SO₂ are the primary decomposition products with comparable concentrations, which aligns with the key predictions of the computational model. Furthermore, the observed consistent increase in their yields with rising temperature, extended reaction time, and elevated pressure firmly establishes the validity of the proposed decomposition mechanism."

35. *In line 424, the authors should clarify which predictions of the simulation they mean.*

That concentrations of CF₄ and SO₂ are "comparable", or that CF₄ and SO₂ are the main products, or something else?

Response 35:

We thank the reviewer for requesting this clarification. The sentence has been revised to specify the precise predictions confirmed by the experiment:

Page 20, line 1. "As shown in **Figure 5e**, the quantitative results demonstrate that CF₄ and SO₂ are the primary decomposition products with comparable concentrations, which aligns with the key predictions of the computational model. Furthermore, the observed gradual increase in their yields with rising temperature, extended reaction time, and elevated pressure firmly establishes the validity of the proposed decomposition mechanism."

36. In the conclusion: the authors should rewrite the conclusion of the influence of pressure (lines 447ff), clarifying also which reaction rate they mean (line 450).

Response 36:

We thank the reviewer for pointing out the lack of clarity in this concluding paragraph. We have rewritten it to precisely articulate the influence of pressure and mixture composition:

Page 21, line 10. "Furthermore, a higher initial concentration of CF₃SO₂F increases the absolute number of decomposition events. The accelerated accumulation of primary products such as CF₄ and SO₂ favors their own secondary reactions. The initial decomposition rate of CF₃SO₂F is significantly promoted by elevated system pressure due to the corresponding increase in molecular collision frequency."

37. Typos and language (incompleteness cannot be ruled out):

line 59: ... 1.3~1.6 that of SF₆ ...

line 74: hampers impedes

line 101: ...of a cf₃so₂f gas mixture... or ...of cf₃so₂f mixtures...

line 103: ...effects of the CF₃SO₂F...

line 108: ...aiming to characterize...

Fig.1 caption: ...calculated with different conditions.

Tab.1 caption: ... of the ... systems.

line 221: ...is shown... or The concentration [profiles] ... are shown...

line 224: raises -> raising

line 231: molecular -> molecule numbers

line 234: ...the increased... ...the decomposition... ...of the...

line 241: possess -> possessing

line 269: ...temperature increases...

line 273: decomposes -> decompose x2

line 324: past -> present tense

line 371: past -> present tense

line 419: ...stable and ?...

Author contributions: methodolog -> methodology

Response 37:

We thank the reviewer for their meticulous review and for identifying these typographical and grammatical errors. We have carefully corrected all the instances highlighted in the review, as listed below:

- Page 3, line 17 "... 1.3~1.6 times of SF₆ ..." has been changed to "... 1.3~1.6 times that of SF₆ ..."
- Page 4, line 3 "hampers impedes" has been corrected to "impedes"
- Page 4, line 8 "...of CF₃SO₂F gas mixture..." has been changed to "... of CF₃SO₂F mixtures..."
- Page 5, line 5 "... effects of CF₃SO₂F..." has been clarified to "... effects of the CF₃SO₂F..."
- Page 5, line 11 "... aiming to characterizing..." has been revised to "...aiming to characterize..."
- Fig. 1 Caption: "... CF₃SO₂F decomposition products at different temperatures calculating." has been rephrased to "... simulating the decomposition of CF₃SO₂F with different conditions."
- Tab. 1 Caption: "... of CF₃SO₂F/CO₂ system." has been completed to "... of the

simulated CF₃SO₂F/CO₂ systems."

- Page 10, line 26 "... are shown..." has been changed to "...is shown..."
- Page 10, line 29 "raises" has been corrected to "raising"
- Page 12, line 3. "molecular" has been changed to "molecules"
- Page 12, line 7. "...increased... ..decomposition... ..of..." has been changed to "...the increased... ..the decomposition... ..of the..."
- Page 12, line 18. We have rephrased the sentence that used "possess" to enhance its clarity and precision. "...exceeds the activation barrier for decomposition..."
- Page 14, line 8. "...temperature up..." has been changed to "...temperature increases..."
- Page 14, line 12."decomposes" has been corrected to "decompose" in both instances.
- Page 16, line 15 & Page 18, line 6: The verb tenses have been corrected to the present tense.
- Page 19, line 29. "...stable and ..." has been changed to "... the dominant decomposition products..."
- Author Contributions: "methodolog" has been corrected to "methodology".

We have also performed a full grammatical check of the manuscript to ensure a high standard of written English.

We are grateful to the reviewers for their valuable time and detailed review comments. Your feedback has not only enhanced the quality of this paper but also positively influenced the direction of our future research. Thank you for your valuable support and time!